# Anti-tumor efficacy of a novel CLK inhibitor via targeting RNA splicing and MYC-dependent vulnerability

Kenichi Iwai[1], Masahiro Yaguchi[1], Kazuho Nishimura[1], Yukiko Yamamoto[1], Toshiya Tamura[1], Daisuke Nakata[1], Ryo Dairiki[1], Yoichi Kawakita[1], Ryo Mizojiri[1], Yoshiteru Ito[1], Moriteru Asano[1], Hironobu Maezaki[1], Yusuke Nakayama[2], Misato Kaishima[2], Kozo Hayashi[2], Mika Teratani[2], Shuichi Miyakawa[3], Misa Iwatani[3], Maki Miyamoto[4], Michael G Klein[5], Wes Lane[5], Gyorgy Snell[5], Richard Tjhen[5], Xingyue He[6], Sai Pulukuri[6] & Toshiyuki Nomura[1,*] (ID)

## Abstract

The modulation of pre-mRNA splicing is proposed as an attractive anti-neoplastic strategy, especially for the cancers that exhibit aberrant pre-mRNA splicing. Here, we discovered that T-025 functions as an orally available and potent inhibitor of Cdc2-like kinases (CLKs), evolutionarily conserved kinases that facilitate exon recognition in the splicing machinery. Treatment with T-025 reduced CLK-dependent phosphorylation, resulting in the induction of skipped exons, cell death, and growth suppression *in vitro* and *in vivo*. Further, through growth inhibitory characterization, we identified high CLK2 expression or *MYC* amplification as a sensitive-associated biomarker of T-025. Mechanistically, the level of CLK2 expression correlated with the magnitude of global skipped exons in response to T-025 treatment. MYC activation, which altered pre-mRNA splicing without the transcriptional regulation of CLKs, rendered cancer cells vulnerable to CLK inhibitors with synergistic cell death. Finally, we demonstrated *in vivo* anti-tumor efficacy of T-025 in an allograft model of spontaneous, MYC-driven breast cancer, at well-tolerated dosage. Collectively, our results suggest that the novel CLK inhibitor could have therapeutic benefits, especially for MYC-driven cancer patients.

**Keywords** alternative splicing; Cdc2-like kinase inhibitor; CLK2; MYC
**Subject Categories** Cancer; Pharmacology & Drug Discovery

See also: **F Salvador & RR Gomis** (June 2018)

## Introduction

Pre-mRNA splicing represents a critically important step for various processes such as development, differentiation, and disease (Chabot & Shkreta, 2016; Inoue *et al*, 2016; Vuong *et al*, 2016). Recently, there has been an increased interest in mutations of pre-mRNA splicing factors and their roles on oncogenesis in hematological cancers (Yoshida *et al*, 2011; Dvinge *et al*, 2016). High-frequency mutations of *SF3B1* or *SRSF2* have been described in patients with myelodysplastic syndromes (MDS), chronic myelomonocytic leukemia, and acute myeloid leukemia (AML) (Meggendorfer *et al*, 2012; Donaires *et al*, 2016; Papaemmanuil *et al*, 2016). In addition, mutations in splicing-related genes have also been found in various solid cancers, including lung, breast, and pancreatic cancers (Dvinge *et al*, 2016). In parallel with clinical observations, the attractiveness of the pharmacological modulation of pre-mRNA splicing as a cancer therapy strategy has also increased (Bonnal *et al*, 2012; Lee & Abdel-Wahab, 2016; Salton & Misteli, 2016).

Members of the evolutionarily conserved Cdc2-like kinase (CLK) family, which comprises CLK1–4, play biologically important roles in pre-mRNA splicing by regulating serine–arginine-rich (SR) proteins. Upon phosphorylation by CLK, SR proteins relocate from nuclear speckles to the spliceosome, where they interact with pre-mRNA to facilitate exon recognition in the splicing machinery (Colwill *et al*, 1996; Ghosh & Adams, 2011; Corkery *et al*, 2015). Both the RNAi-mediated depletion of CLK and chemical inhibition of CLK modulate alternative splicing (AS), particularly the skipped exon (SE) type of AS, resulting in the suppression of cell proliferation (Muraki *et al*, 2004; Fedorov *et al*, 2011; Dominguez *et al*, 2016; Sako *et al*, 2017).

1 Oncology Drug Discovery Unit, Takeda Pharmaceutical Company, Limited, Fujisawa, Japan
2 Integrated Technology Research Laboratories, Takeda Pharmaceutical Company, Limited, Fujisawa, Japan
3 Biomolecular Research Laboratories, Takeda Pharmaceutical Company, Limited, Fujisawa, Japan
4 Drug Metabolism & Pharmacokinetics Research Laboratories, Takeda Pharmaceutical Company, Limited, Fujisawa, Japan
5 Department of Structural Biology, Takeda California Inc., San Diego, CA, USA
6 Oncology Drug Discovery Unit, Takeda Pharmaceuticals International Co., Cambridge, MA, USA
*Corresponding author. Tel: +81-466-32-1948; Fax: +81-466-29-4461; E-mail: toshiyuki.nomura@takeda.com

Recently, observations that the well-known proto-oncogene MYC controls pre-mRNA splicing and that MYC-driven cancers are susceptible to spliceosome inhibition have highlighted the use of pharmacological splicing modulators as promising anti-cancer agents for MYC-driven cancers (Hsu *et al*, 2015; Koh *et al*, 2015). Aberrant MYC activation, mediated by *MYC* translocation, amplification, and mutation, is a frequent event in various hematological and solid cancers (Dang, 2012; Kress *et al*, 2015). Although numerous studies have successfully targeted these cancers, MYC remains a highly significant therapeutic target (Delmore *et al*, 2011; Kessler *et al*, 2012; Cermelli *et al*, 2014; Camarda *et al*, 2016; Horiuchi *et al*, 2016).

Here, we hypothesized that CLK inhibition might function as a novel pre-mRNA splicing modulation-based anti-cancer strategy, especially for MYC-driven cancers. Our findings, which demonstrate the ability of an orally available CLK inhibitor to effectively target MYC-driven cancers, address a novel biological interaction of CLK inhibition with MYC activation.

## Results

### T-025 is a highly potent CLK inhibitor

To investigate an anti-tumor efficacy of a CLK inhibitor in animal models, we developed a new class of CLK inhibitors. Specifically, we chemically modified a 7*H*-pyrrolo[2,3-*d*]pyrimidine derivative by measuring the CLK2 inhibitory activity and determining the co-crystal structure with CLK2. After optimizing the lead compound, we discovered an orally available and highly potent CLK2 inhibitor, T-025 ($N^2$-methyl-$N^4$-[pyrimidin-2-ylmethyl]-5-[quinolin-6-yl]-7*H*-pyrrolo[2,3-*d*]pyrimidine-2,4-diamine; Fig 1A). The co-crystal structural analysis revealed that T-025 inserts into the CLK2 ATP-binding site and interacts with Glu244 and Leu246 in the CLK2 hinge region (Fig 1B). However, a KINOME*Scan*-based kinase selectivity evaluation identified T-025 as a highly selective inhibitor of CLK ($K_d$ values to CLK1, CLK2, CLK3, and CLK4 were 4.8, 0.096, 6.5, and 0.61 nmol/l, respectively) and dual-specificity tyrosine-phosphorylation-regulated kinase 1 (DYRK1) family proteins ($K_d$ values to DYRK1 and DYRK1B were 0.074 and 1.5 nmol/l, respectively) (Fig 1C). No other kinases outside of the DYRK1 family had $K_d$ values < 30 nmol/l, suggesting that T-025 is a potent CLK/DYRK1 inhibitor with > 300-fold enhanced selectivity for these kinases than to other kinases.

### T-025 induced skipping exon, resulting in anti-proliferative effect in MDA-MB-468 *in vitro* and *in vivo*

Our previous report shows that CLK inhibitors suppress cell proliferation and induce cell death in MDA-MB-468 cells, accompanied by several CLK-associated downstream effects including a global modulation of AS events (Araki *et al*, 2015). In line with our previous report, treatment of MDA-MB-468 cells with T-025 suppressed the phosphorylation of SR protein detected with 1H4 monoclonal antibody (Fig 2A), resulting in growth inhibition (Fig 2B) with apoptosis as detected by an increase in the sub-G1 population (fluorescence-activated cell sorting analysis; Appendix Fig S1A). To further assess the *in vitro* cellular inhibition of CLK, we generated a

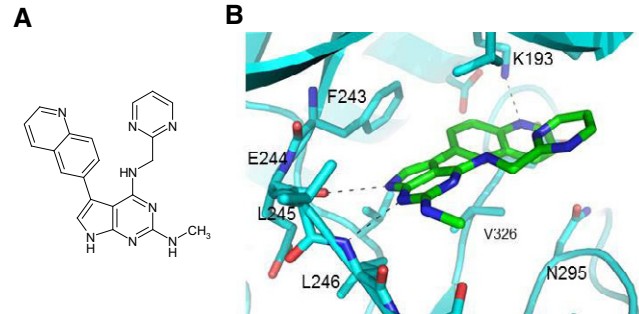

**A**

**B**

**C**

| | | $K_d$ (nmol/L) | % inhibition (300 nmol/L) |
|---|---|---|---|
| CLK | CLK1 | 4.8 | 99.4 |
| | CLK2 | 0.096 | 93.1 |
| | CLK3 | 6.5 | 90.2 |
| | CLK4 | 0.61 | 100 |
| DYRK | DYRK1A | 0.074 | 100 |
| | DYRK1B | 1.5 | 99.6 |
| | DYRK2 | 32 | 98.6 |
| HIPK | HIPK1 | 55 | 94.6 |
| | HIPK2 | 96 | 96.0 |
| | HIPK3 | N.T. | 95.1 |
| | HIPK4 | N.T. | 95.6 |
| | YSK4 | 33 | 97.5 |
| | IRAK4 | 54 | 94.8 |
| | FLT3 D835H | 94 | 98.8 |
| | FLT3 D835Y | N.T. | 93.3 |
| | ERK8 | 140 | 94.8 |
| 440 kinases | | N. T. | < 90 |

**Figure 1.  T-025 is a novel and potent CLK inhibitor.**

A   Chemical structure of T-025.

B   X-ray crystal structure of CLK2 with T-025 (see also Appendix Fig S7 for detailed co-crystal structure data).

C   $K_d$ values of T-025 against CLK and DYRK family kinases and the result of a panel of various kinases. N.T., not tested. Binding activities of T-025 at 300 nmol/l were measured against 468 kinases ($n$ = 2).

new antibody that recognized phosphorylated Ser98 of CLK2 (pCLK2), which is reported as an auto-phosphorylation of CLK2 (Rodgers *et al*, 2010), and our *in vitro* assays also supported this previous finding (Appendix Fig S2A). Immunoblotting with the pCLK2 antibody revealed treatment with T-025 decreased both pCLK2 and CLK2 (Fig 2A), and quantified band intensities showed relative phosphorylation level was reduced in a dose-dependent manner (Appendix Fig S1B). Considering with a previous finding that kinase activity of CLK2 contributed to stability of CLK2 protein (Rodgers *et al*, 2010), our result suggested that T-025 inhibited the kinase activity of CLK2 in cultured MDA-MB-468 cells, leading to the degradation of CLK2.

We next evaluated a CLK inhibition-mediated AS. As for a specific CLK inhibition-dependent AS event, we confirmed that T-025 induced the skipping exon 7 of *RPS6KB1* (Appendix Fig S1C), which is also induced by other CLK inhibitors and RNAi-mediated depletion of CLK2 (Araki *et al*, 2015), followed by a reduction in the protein level of S6K (Appendix Fig S1D). A whole transcriptome RNA sequencing (RNA-Seq) and its consequent splicing analysis using a mixture of isoforms (MISO) (Katz *et al*, 2010) revealed that T-025 at the concentrations of 30, 100, and 300 nmol/l

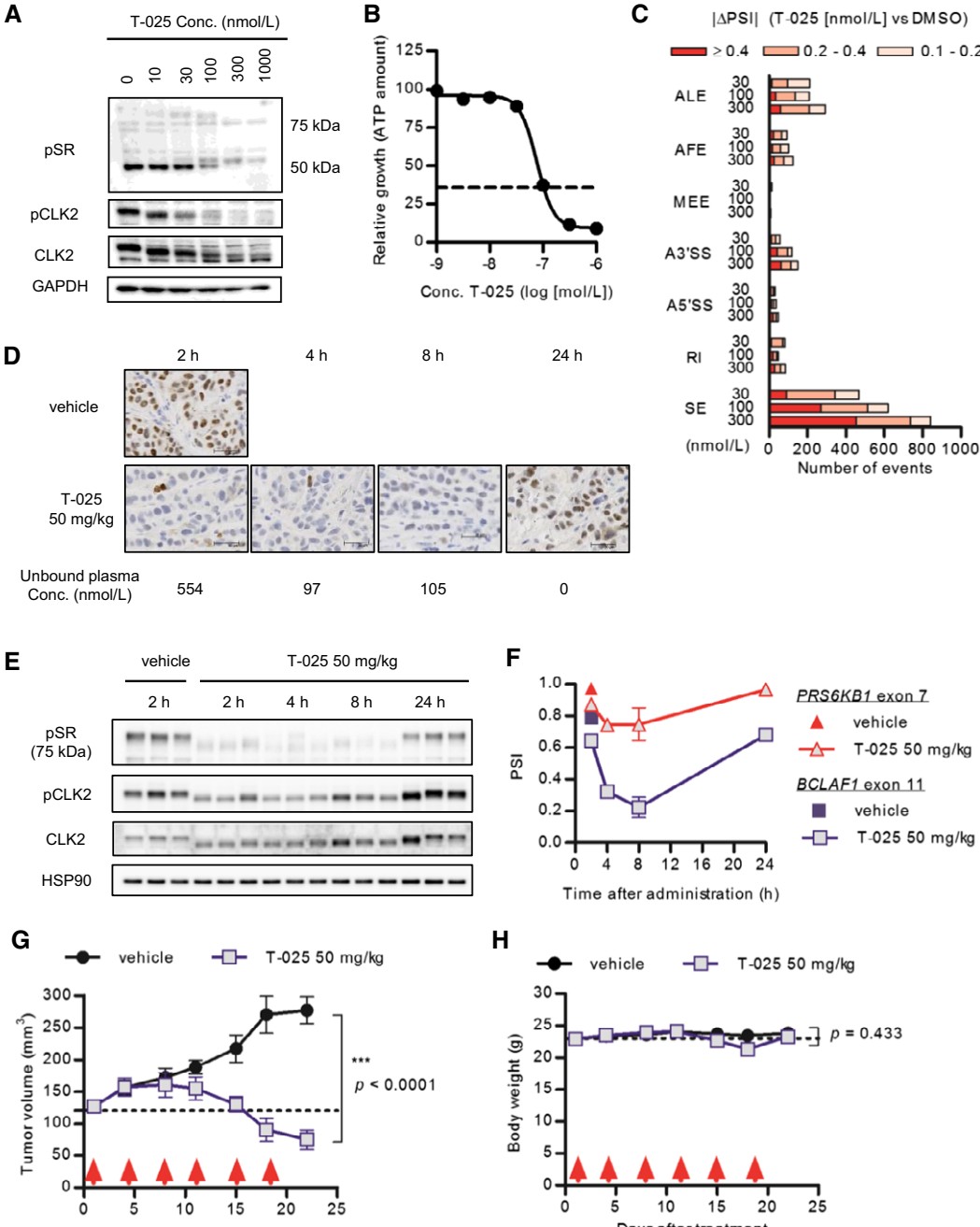

**Figure 2. T-025 exhibited anti-tumor efficacy in MDA-MB-468 xenografts.**

A    MDA-MB-468 cells were treated with T-025 for 6 h, and phosphorylation levels were detected via immunoblotting with phospho-specific antibody.

B    T-025 dose–response curve in MDA-MB-468 cells for 72-h treatment. The black dotted line indicates the relative ATP value prior to treatment (Day 0).

C    The number of AS events modulated by T-025 treatment for 6 h in MDA-MB-468 cells. The numbers of AS events with a BF > 20 and |ΔPSI| > 0.1 were counted and categorized according to the AS type (SE, skipped exon; RI, retained intron; A5′SS, alternative 5′-splice-site; A3′SS, alternative 3′-splice-site; MEE, mutually exclusive exon; AFE, alternative first exon; ALE, alternative last exon).

D–F    MDA-MB-468 xenograft tumors treated with 50 mg/kg of T-025 were sampled and analyzed by immunohistochemistry (F), immunoblotting (E), or RT–PCR (F). Representative images of pCLK2 stained tumors and scale bar (30 μm) are shown. Also shown: The mean unbound plasma T-025 concentration was calculated using protein binding after oral administration.

G, H    Anti-tumor efficacy of T-025 in MDA-MB-468 xenograft models. T-025 was administered twice daily on 2 days per week (red arrows). Tumor volume (G) and body weight (H) during the efficacy study are shown as means ± s.e.m. (*n* = 5).

Data information: In (B, D, and F), data are shown as the means ± s.d. of three independent experiments (*n* = 3). In (G and H), an unpaired Student's *t*-test was performed.

Source data are available online for this figure.

(approximate $IC_{50}$ values for growth inhibition) largely modulated AS via SE in a dose-dependent manner (Fig 2C). Further, when we carefully screened the observed AS events, we found that the skipping of exon 11 of *BCLAF1*, as an additional downstream AS event, was one of the most sensitive and largest events among the alternative SEs (Appendix Fig S1E). Together, these results in cultured MDA-MB-468 cells indicated that T-025-induced cell death, accompanied by the phenotypes that are previously observed by other CLK inhibitors or RNAi-mediated depletion.

Then, we evaluated T-025 in an animal model. The pharmacokinetics evaluation of T-025 in nude mice revealed that the unbound plasma concentrations of T-025 were 554, 97, and 104 nmol/l at 2, 4, and 8 h, respectively, following the oral administration of T-025 at 50 mg/kg (Fig 2D); these concentrations were sufficient to suppress the CLK-dependent phosphorylation and to induce skipping exon in various genes including exon 7 of the *RPS6KB1* (Fig 2C and Appendix Fig S1C). Therefore, we performed a pharmacodynamics assessment of T-025 at 50 mg/kg in MDA-MB-468 xenograft tumors, and found that pCLK2 detected with immunohistochemistry and immunoblotting decreased from 2 to 8 h after oral administration (Fig 2D and E), followed by a reduction in the *RPS6KB1* exon 7 and *BCLAF1* exon 11 percentage splice-in (PSI) values (Fig 2F).

An efficacy study in a MDA-MB-468 xenograft model was performed with a regimen of twice daily on 2 days per week schedule. The treatment yielded profound anti-tumor effects, illustrating that the tumor volumes had shrunk relative to the initial volumes at the end of the 3-week treatment cycle (Fig 2G). Additionally, although the T-025 dosage was near the maximum tolerated dose, it was apparently well tolerated with a < 10% nadir body weight loss (Fig 2H). Taken together, these results using MDA-MB-468 xenografts suggested T-025 had an anti-tumor efficacy at tolerable dosage, accompanied by the modulation of downstream markers.

## Solid cancer cell lines harboring *MYC* amplification or high CLK2 expression were more sensitive to T-025

For the characterization of T-025 as an anti-tumor agent, we subjected T-025 to a panel of growth inhibition assays in 240 cancer cell lines and a subsequent unbiased bioinformatics analysis by utilizing OncoPanel 240. Consequently, T-025 exerted a broad range of anti-proliferative activities in both hematological and solid cancer cell lines ($IC_{50}$ values: 30–300 nmol/l), sensitivity to this drug was not organ of origin- or disease type-dependent (Fig 3A). The unbiased bioinformatics analysis flagged several biomarker candidates that were significantly associated with sensitivity; analysis of mRNA expressions identified genes that were significantly expressed higher/lower in the top 25% sensitive cancer cell lines than in the bottom 25% cancer cell lines (Fig EV1A). In the sensitivity-associated mRNAs, we found that the expression of CLK2 was significantly higher in the sensitive cell lines with a *P*-value of 1.58E-09, which is much lower than the *P*-value from other CLK family or DYRK family. Considering the primary target of T-025 as well as the oncogenic role of CLK2 in breast cancer (Yoshida *et al*, 2015), we hypothesized that cancer cells with higher CLK2 expression were dependent on the CLK2 kinase activity for their survival. Another unbiased analysis using mutations and copy number alterations identified 14 statistically significant biomarker candidates, including amplified- or mutated-*MYC* (Fig EV1B). Recent reports that

spliceosome inhibition is more effective against MYC-driven cancer (Hsu *et al*, 2015; Koh *et al*, 2015) persuaded us to validate this preliminary analysis result that *MYC*-mutated or *MYC*-amplified cancer cell lines were more sensitive.

We further analyzed 169 (19 hematological and 150 solid cancer) cell lines of 240 cell lines, whose genomic data such as expression, mutation, and copy number variation (CNV) could be obtained from the Cancer Cell Line Encyclopedia (CCLE) database (Barretina *et al*, 2012; Fig EV2A). With a careful evaluation of the *MYC* genetic status to include the role of mutation and to remove passenger mutations, we found that solid cancer cell lines exhibiting *MYC* alteration (only *MYC*-amplified cell lines were found) were significantly more sensitive to T-025 than other solid cancer cell lines ($P = 0.0042$, Fig 3B). Conversely, in the 19 hematological cancer cell lines, we did not observe higher sensitivity associated with *MYC* alterations (amplified, driver-mutated, translocated; Fig EV2B and C). Since other MYC family proto-oncogenes such as N-Myc or L-Myc share several functions with MYC (Malynn *et al*, 2000), we additionally considered the gene status of other MYC family; however, we could not find any sensitivity correlating with MYC family gene alteration in hematological cancer cell lines (Fig EV2D). Interestingly, solid cancer cell lines with amplified MYC family genes, such as *MYC*, *MYCN*, or *MYCL*, were more sensitive than those without amplified MYC family genes with *P*-value at 0.0010 (Fig EV2E), suggesting that common downstream effects of MYC family genes are involved in the sensitivity to T-025. Notably, we found that half of the top 20% sensitive solid cancer cell lines (15 out of 30 cell lines) harbored amplified MYC family genes. Regarding the expression of CLK2, when we divided solid or hematological cancer cell lines into three groups on the basis of CLK2 expression (high, medium, and low), solid cancer cell lines with high CLK2 expression were significantly more sensitive than those with low CLK2 expression ($P < 0.0001$, Fig 3C), but not hematological cancer cell lines with high CLK2 expression (Fig EV2F). In summary, these analyses revealed that the expression of CLK2 or amplified *MYC* was statistically associated with the sensitivity to T-025 in the solid cancer cell lines.

## T-025 modulated AS with a magnitude depending on CLK2 expression

To evaluate the hypothesis generated from the bioinformatics analysis, we first assessed the protein level of CLK2 in 56 various cancer cell lines because the protein level of CLK2 is also regulated by ubiquitination-dependent degradation (Bidinosti *et al*, 2016). The findings of the immunoblot analysis of CLK2 revealed that the protein levels of cell lines had a large variation, and MDA-MB-468 cells appeared to express a relatively high CLK2 protein level (11[th] in 56 cell lines, Appendix Fig S3); furthermore, the protein expression level of CLK2 significantly correlated with sensitivity (Fig 4A). In addition, we found that T-025 showed high *in vitro* growth suppressive effect in an additional cancer cell line with higher CLK2 protein, that is, lung cancer NCI-H1048; also, T-025 caused moderate anti-proliferative effect in normal fibroblast cell lines with lower CLK2 protein (Fig 4B). Further, treatment with T-025 exhibited *in vivo* anti-tumor efficacy in NCI-H1048 xenografts (Fig EV3A), with no significant body weight loss (Fig EV3B). In contrast, although the CLK2 protein was not associated with sensitivity to T-025 in

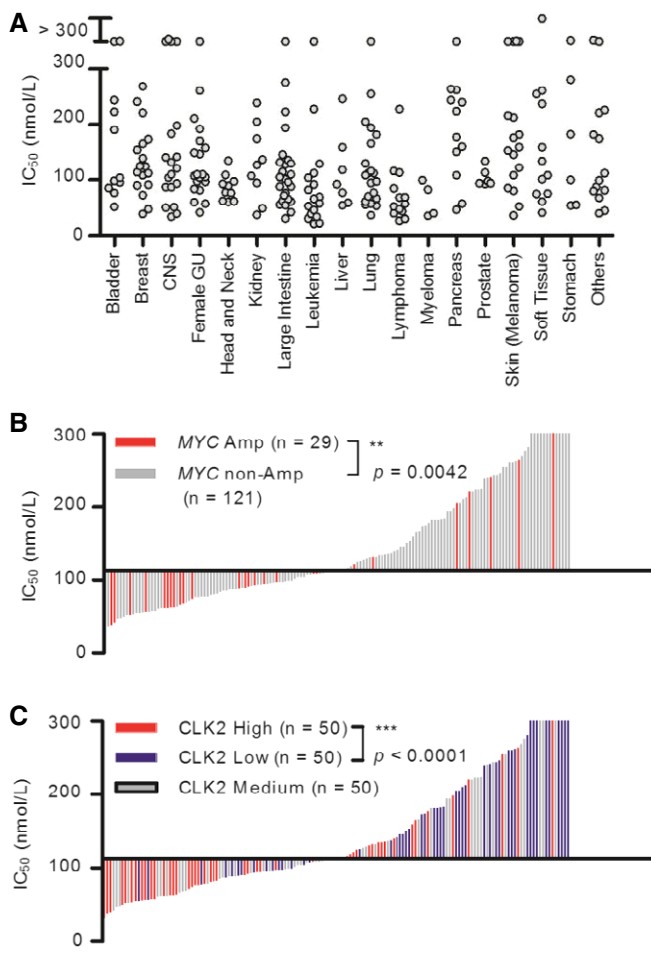

**Figure 3.  T-025 exhibited a board range of anti-proliferative effect in a panel of cancer cell lines.**

A   $IC_{50}$ values of T-025 in 240 cell lines. Each gray circle indicates a single cell line sorted according to its original organ/disease type.
B   Correlation between T-025 sensitivity and *MYC* amplification status in solid cancer cell lines (*n* = 150). Each bar indicates a single cell line, and red bar indicate cell lines with amplified *MYC*.
C   Correlation between T-025 sensitivity and CLK2 expression status in solid cancer cell lines (*n* = 150). Each bar indicates a single cell line, and red, gray, or blue bar indicate cell lines with high, medium, or low CLK2.

Data information: In (B and C), a Mann–Whitney test was performed.

hematological cancer cell lines (Fig EV3C), we found that T-025 showed profound anti-tumor efficacies in xenografts in an MV-4-11 AML cell line (Fig EV3D), which was one of the most sensitive hematological cancer cell lines in the panel ($IC_{50}$ = 30.1 nmol/l), as well as in xenografts derived from AML patient (PDX) (Fig EV3E).

To investigate how high CLK2 kinase expressions affected sensitivity to T-025, we asserted the AS event caused by T-025 in additional solid cancer cell lines with different CLK2 expression and sensitivity, such as NCI-H1048 lung cancer cells ($IC_{50}$ at 84.4 nmol/l), COLO320HSR colorectal cancer cells with high CLK2 expressions ($IC_{50}$ at 91.8 nmol/l), and 786-O kidney cancer cells with low CLK2 expressions ($IC_{50}$ at 340 nmol/l). The AS analysis of RNA-seq of the additional three cancer cell lines revealed that treatment with T-025 mainly caused SE type of AS events in all tested cell lines (Fig 4C),

and the AS events largely changed by the treatment were similar among all tested cell lines (Fig 4D). Because most AS events modulated by T-025 were independently determined based on CLK2 expression levels, we assessed whether the degree of AS in each cell line was associated with its CLK2 expression levels. The degree change in commonly modulated SE type of AS events was the largest in NCI-H1048 cells and smallest in 786-O cells (Fig 4E), which suggests that the expression level of CLK2 correlated with the degree of AS in response to T-025 treatment. The pathway analysis of genes with commonly modulated SE type of AS events revealed that T-025 modulated AS of the genes in the cell cycle, DNA repair, RNA splicing, and RNA transport pathways (Fig 4F), indicating that T-025 modulated these essential pathways in cancer cells via AS with a magnitude depending on its CLK2 expression level.

### CLK inhibition and MYC activation synergistically induced apoptosis

We then questioned how *MYC* amplification connected with the sensitivity to T-025. To examine whether an increased MYC-activity rendered cancer cells more sensitive to T-025, we asserted the effect of MYC activation using SK-MEL-28 cells with doxycycline (Dox)-inducible MYC expression. The Dox-dependent MYC induction was confirmed with an increased MYC mRNA (Fig 5A) and an enhanced downstream transcriptional activity (Fig 5B). SK-MEL-28 cell lines that were pretreated with Dox were subsequently treated with T-025, and MYC activation induced a greater sensitivity to T-025 (Fig 5C), which was accompanied by a T-025 dose-dependent caspase-3/7 activation (Fig 5D) and PARP1 cleavage under MYC-activating conditions (Fig 5E).

To validate further, as well as to rule out the possibility that MYC-dependent enhanced effect was caused by DYRK1A inhibition, we additionally used 4-hydroxytamoxifen (4OHT)-inducible MYC nuclear translocating/activating U2OS cells (Fig 5F), whose CNV and expression of DYRK1A was one of the lowest in the CCLE cell lines (Log2 CNV = −1.55, expression *Z*-Score = −2.65, Appendix Fig S4A). As a result, treatment with T-025 exhibited higher growth suppressive effect under MYC activation (Fig 5G), accompanied by cPARP1 induction (Fig 5H). In addition, we found that the RNAi-mediated downregulation of MYC reduced the growth suppressive effect of T-025 in *MYC*-amplified SK-BR3 or MCF7 breast cancer cell lines, whereas the depletion of DYRK1A did not reduce the effect of T-025 (Fig 5I and Appendix Fig S4B). Altogether, these results suggest that MYC activation rendered cancer cells vulnerable to T-025-mediated CLK inhibition.

We demonstrated that T-025 caused common alternative SEs in cancer cell lines and showed that there was more effective growth inhibition in cancer cell lines with high CLK2 expression or with MYC activation. To determine whether these effects were mainly attributable to CLK inhibition and applicable for other CLK inhibitors, we assessed another CLK inhibitor (T3), which differed from T-025 with respect to its scaffold structure and sensitivity to the DYRK1 family of kinases (Fig EV4A and B; Funnell *et al*, 2017). First, we tested whether 125 SEs, which were commonly modulated by T-025 at 300 nmol/l in four cancer cell lines (Fig 4E), were also modulated in the HCT116 cells treated with T3. Consequently, SEs mostly changed in HCT116 cells treated with T3: 58.4% (73 out of 125) at 500 nmol/l and 77.6% (97 out of 125) at 1,000 nmol/l

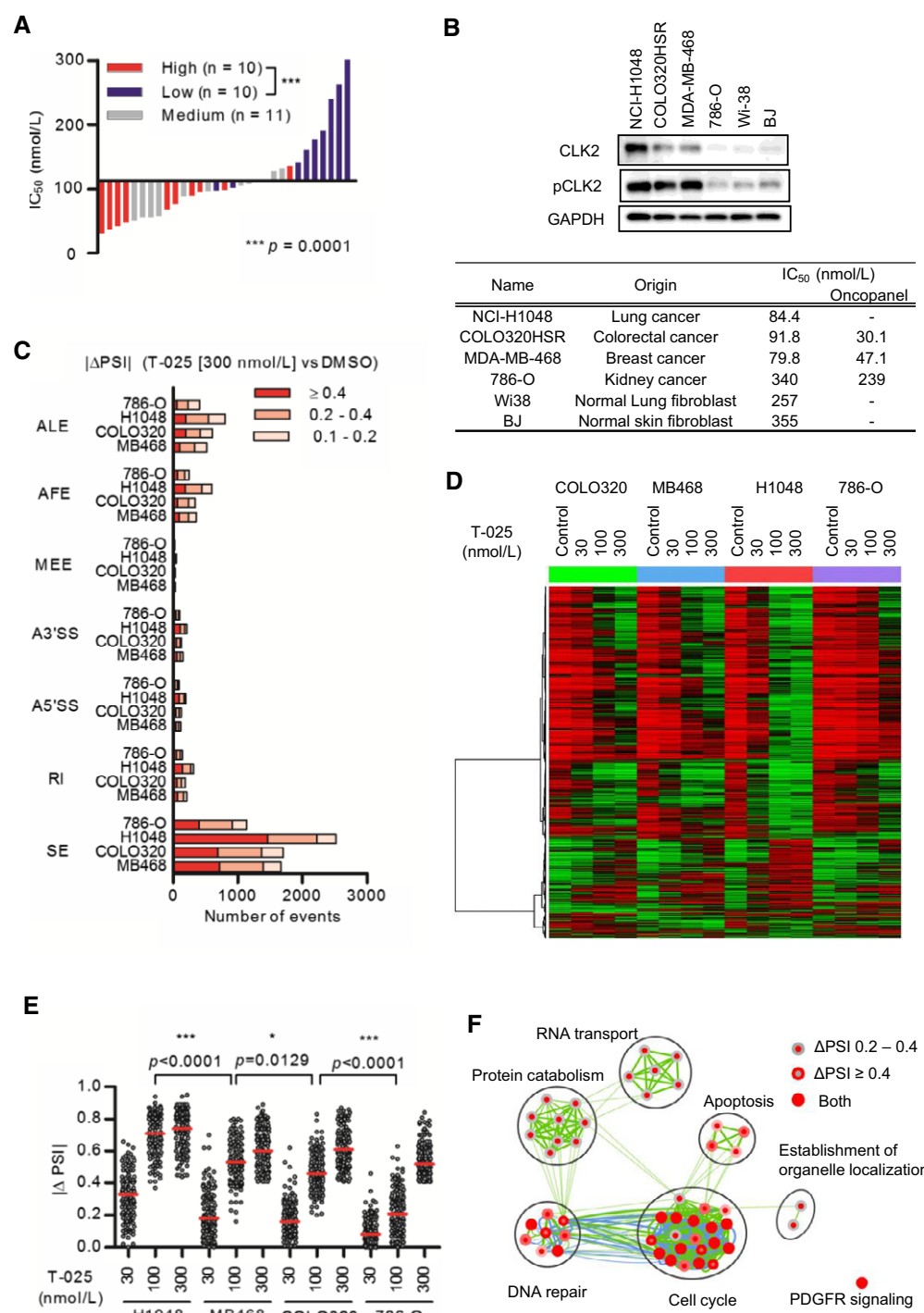

**Figure 4. T-025 modulated AS with a magnitude depending on CLK2 expression.**

A  Correlation between growth suppressive sensitivity to T-025 and the protein level of CLK2 in solid cancer cell lines. The CLK2 band intensity of each cell line was normalized by that of HCT116 and 786-O, and divided into three groups (high, medium, and low). A Mann–Whitney test was performed.

B  The $IC_{50}$ values of T-025 treatment for 72 h and expression level of CLK2 and pCLK2 in representative cancer cell and normal fibroblast lines.

C  The number of AS events modulated by T-025 treatment for 6 h in NCI-H1048 (H1048), COLO320HSR (COLO320), and 786-O, in addition to MDA-MB-468 (MB468) cells (Fig 2C). The numbers of AS splicing events with a BF > 20 and $|\Delta PSI| > 0.1$ were counted and categorized according to the AS type.

D  AS events with $|\Delta PSI| > 0.4$ at 300 nmol/l were clustered and described as a heat map.

E  $|\Delta PSI|$ values of T-025 dependent SEs, that commonly modulated with a BF > 20 and $|\Delta PSI| > 0.4$ by the T-025 at 300 nmol/l in the four cell lines, were described (n = 125). A Steel–Dwass test was performed.

F  Pathways enriched by genes of T-025 dependent SEs were analyzed with CytoScape and described.

Source data are available online for this figure.

(Fig EV4C). Second, we evaluated T3 in a panel of 60 solid cancer cell lines and found that the growth inhibition profile of T3 was similar to that of T-025 ($r^2$ = 0.783, Fig EV4D), and cancer cell lines with a high CLK2 protein level also showed a higher growth inhibitory response to T3 (Fig EV4E). Finally, we found T3 showed a stronger growth suppressive effect on SK-MEL-28 cells concomitant with MYC activation (Fig EV4F). Collectively, T3 exhibiting similar profiles as T-025 suggested that these effects caused by inhibitors were mainly due to CLK inhibition.

### MYC independently regulated AS on transcriptional regulation of CLK

Next, we investigate the molecular connection between MYC and CLKs. MYC has been reported to modulate AS through the transcriptional regulation of SRSF1 (Das *et al*, 2012) and core-spliceosome components, including PRMT5 (Koh *et al*, 2015). Because we found that the solid cancer cell lines with amplified *MYC* showed significantly higher expression of CLK2 in the 150 solid cancer cell lines we analyzed (Appendix Fig S5A), we tested whether MYC also transcriptionally regulates CLK2 and other CLK family kinases. In the MYC-inducible SK-MEL-28 cells, the Dox-dependent MYC induction upregulated the mRNA expression of PRMT5 and SRSF1. However, we did not observe an upregulation of CLK2 or other CLK family kinases (Fig 6A), suggesting that CLK family kinases were not direct transcriptional targets of MYC. In addition, increased protein level of CLK2 was not observed in the SK-MEL-28 and U2OS cells with MYC induction (Appendix Fig S5B).

We also performed a RNA-Seq to observe the MYC induction-dependent global transcriptional change as well as the MYC induction-dependent global modulation of AS. Consistent with our RT–PCR results, we found that the MYC induction increased several splicing-related genes, but not the CLK family kinases (Fig 6B). Splicing analysis of the RNA-Seq data confirmed that MYC induction caused AS, which indicated that although Dox treatment modulated some AS events in control SK-MEL-28 cells, extensive modulation occurred in MYC-inducible cells (Fig 6C). Since MYC regulated AS without the transcriptional regulation of CLKs, we hypothesized that AS caused by MYC induction and that caused by CLK inhibitor were exclusive. When we compared AS events commonly modulated by T-025 in the four cancer cell lines with MYC-dependent AS events, only five of the 546 events (|ΔPSI| > 0.2) associated with T-025 treatment were observed among MYC-dependent AS events (Fig 6D). Together, these results suggested that MYC transcriptionally regulated various splicing-related genes, but not CLKs, resulting in the MYC-dependent modulation of AS that differed from those modulated by the CLK inhibitor.

We next examined the effect of T-025 in the cells under MYC induction. In line with the hypothesis that protein levels of CLK2 determine the degree of AS, treatment of MYC-inducible SK-MEL-28 cells with T-025 reduced pCLK2 and CLK2, resulting in dephosphorylation of SR proteins, skipped exon 7 of *RPS6KB1*, and reduced protein level of S6K in both MYC-induced and non-MYC-induced conditions at equivalent levels (Fig 6E and Appendix Fig S5C). Then, we asked whether T-025 modulated AS of the transcriptional target gene of MYC, accompanied by suppression of the MYC downstream target. We found that T-025 caused the skipping of exon 3 of *NOP16* (also known *HSPC111*), whose transcription is regulated by

MYC (Butt *et al*, 2008), followed by the induction of a flame-shift type of transcriptional variant. The treatment of T-025 caused the AS of *NOP16* in MYC-activated condition with equivalent degree to non-MYC-activated conditions (Fig 6F), resulting in the suppression of MYC-dependent upregulation exon 3 including *NOP16* variant (Fig 6G). This suggested that T-025 suppressed the upregulation of MYC transcriptional target gene *NOP16* through modulation of AS.

### T-025 exhibited significant anti-tumor efficacy in an MYC-driven breast tumor allograft model

We finally evaluated the effects of T-025 in a mouse tumor model with the intent to validate this molecule as a therapy for MYC-driven tumors. We collected MYC-driven spontaneous breast cancer tumors from MMTV-*MYC* transgenic mice and established an allograft model by subcutaneously implanting tumor tissues into nude mice. T-025 strongly suppressed growth of the allograft tumors (Fig 7A) without inducing body weight losses (Fig 7B), suggesting that CLK inhibitors could exert anti-tumor effects against MYC-driven breast cancers.

To apply our hypothesis that CLK2-high and *MYC*-amplified cancers were more sensitive to T-025 in clinical stratification, we re-assessed the result of our growth inhibition panel on the basis of original organ type and evaluated public clinical data. We observed that breast cancer cell lines with both amplified *MYC* and highly expressed CLK2 were significantly sensitive to other breast cancer cell lines (Fig EV5); however, we did not observe significant difference in cancer cell lines from other original organ types. Importantly, although CLK2 expression level was independent of *MYC* amplification in samples from the published clinical breast cancer data (Curtis *et al*, 2012; Pereira *et al*, 2016; Appendix Fig S6A), patients with both amplified *MYC* and high CLK2 expression show poor prognosis, with the lowest medium survival time relative to patients with only amplified *MYC* or those with only highly expressed CLK2 (Appendix Fig S6B). Further, although only amplified *MYC* did not significantly worsen the prognosis in breast cancer patients categorized as ER-positive, HER2-negative, and high-proliferative subtype, patients with both high-expressed CLK2 and *MYC* amplification showed significantly poor outcomes (Fig 7C). These clinical observations suggest that high expression of CLK2 and *MYC* amplification worsen the prognosis of particular subtypes of breast cancer patients and that CLK inhibitor would be more effective to these poor prognosis cancers.

## Discussion

We have discovered an orally available and potent inhibitor of CLK. T-025 inhibited tumor growth in both *in vitro* and *in vivo* models, accompanied by the modulation of pre-mRNA splicing. Our data using T-025 reveal a new therapeutic opportunity for cancer treatment via a novel splicing modulator.

Our *in vitro* and *in vivo* anti-tumor efficacy data of T-025 in various models indicate that the CLK inhibitor exhibits therapeutic potential for cancer treatment. Although T-025 potentially inhibits other kinases, particularly for DYRK1A, we hypothesize that the anti-tumor effects of T-025 were caused via CLK inhibition. First, the different chemical scaffold CLK inhibitor T3 exhibited similar

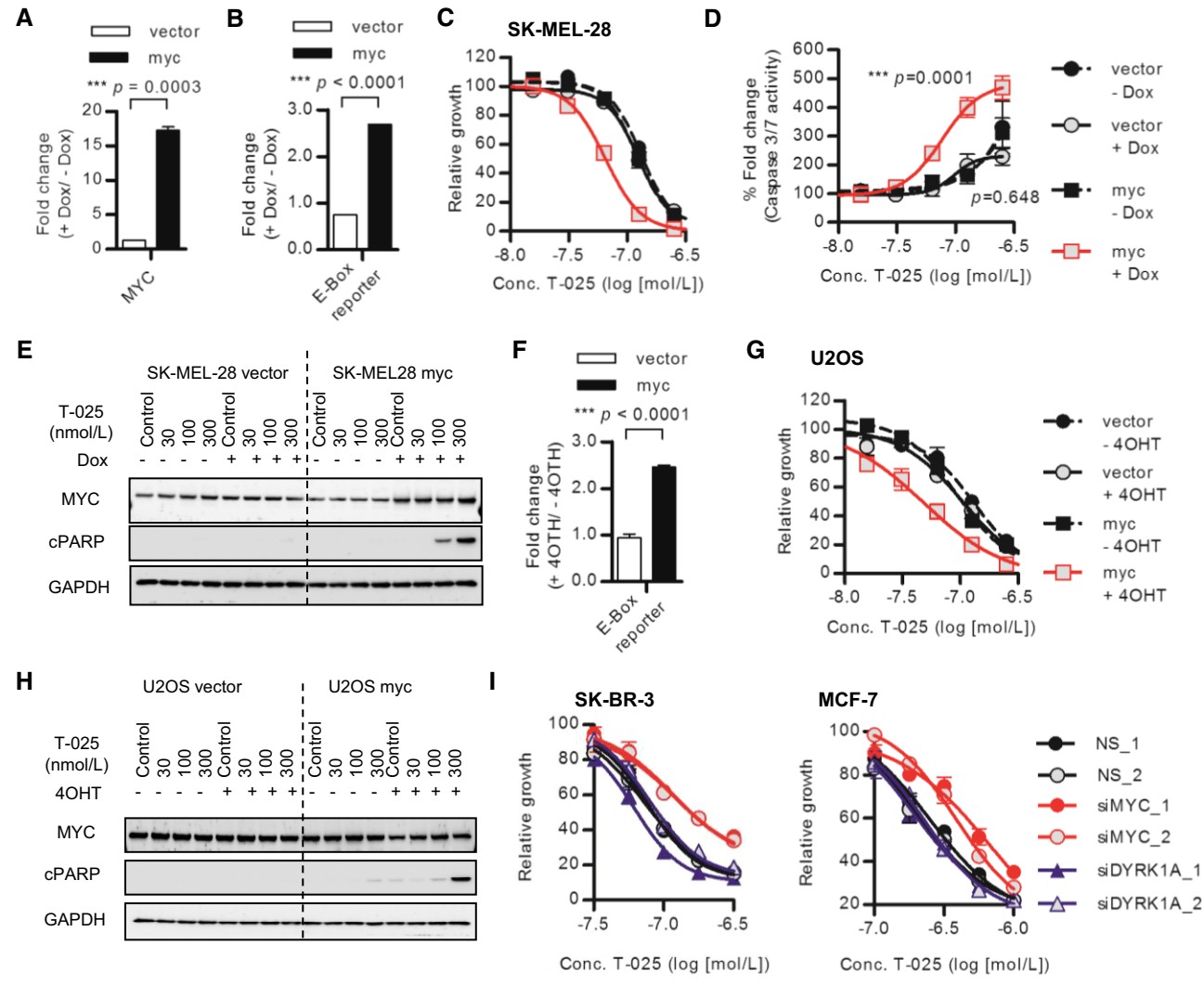

**Figure 5.  MYC activation and T-025 treatment synergistically induced apoptosis.**

A, B     Dox treatment-induced/activated MYC in MYC-inducible SK-MEL-28 cells. SK-MEL-28 cells were treated with 0.5 μg/ml Dox for 54 h, and MYC mRNA level was analyzed with RT–PCR (A). E-Box reporter activities of SK-MEL-28 cells after 48 h of Dox treatment were measured as MYC transcriptional activities.

C         Dose–response growth inhibition curve of T-025 in MYC-inducible SK-MEL-28 cells. SK-MEL-28 cells pretreated with Dox for 48 h were additionally incubated with T-025 for 72 h.

D, E     Apoptosis induction caused by T-025 in MYC-inducible SK-MEL-28 cells. SK-MEL-28 cells pretreated with Dox for 48 h were additionally treated with T-025 for 24 h, and then, apoptosis was measured by caspase-3/7 activity (D) or immunoblotting of cPARP1 (E).

F         4OTH treatment-induced/activated MYC in MYC-inducible U2OS cells. E-Box reporter activities of U2OS cells after 48 h of 4OTH (500 nmol/l) treatment were measured as MYC transcriptional activities.

G         Dose–response growth inhibition curve of T-025 in MYC-inducible U2OS cells. U2OS cells pretreated with 4OTH for 48 h were additionally incubated with T-025 for 72 h.

H         Apoptosis induction caused by T-025 in MYC-inducible U2OS cells. U2OS cells pretreated with 4OTH for 48 h were additionally treated with T-025 for 24 h, and then, apoptosis was measured by immunoblotting of cPARP1.

I         Dose–response growth inhibition curve of T-025 in SK-BR3 or MCF7 cells transfected with MYC or DYRK1A siRNA. The cells pretreated with siRNA for 24 h were additionally incubated with T-025 for 72 h.

Data information: In (A, B and F), an unpaired Student's *t*-test or an unpaired Student's *t*-test with Welch's correction was performed. In (D), Tukey's test was performed. In (A–D, F, G and I), data are shown as the means ± s.d. of three independent experiments (*n* = 3). IC$_{50}$ values and 95% confidence intervals (95% CIs) in (C, G, and I) are described in Appendix Fig S8.

Source data are available online for this figure.

effects to T-025 in terms of AS and the profile of growth inhibition, including MYC induction-dependent higher activity. Notably, AS caused by T3 has been reported to be largely overlapped with that caused by CLK1/2/3/4 siRNA (Funnell *et al*, 2017), suggesting that

AS caused by T-025 was also similar to that caused by the depletion of CLKs. Second, we show that expression level of CLK2 is responsible for the T-025-mediated effects on the induction of AS and the suppression of cell proliferation; The SE type AS caused by T-025

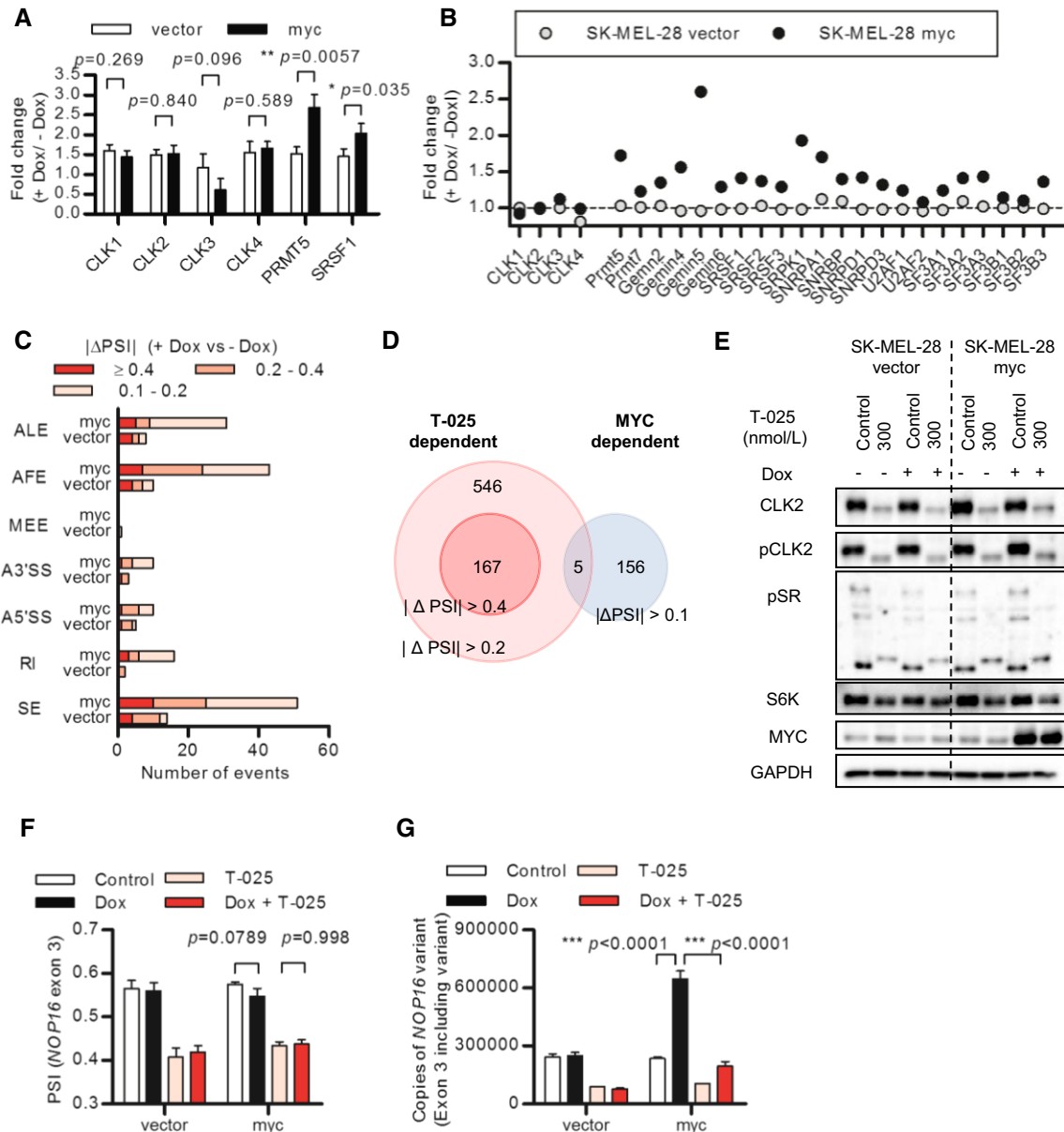

**Figure 6. Ectopic MYC activation altered pre-mRNA splicing without transcriptional regulation of CLK.**

A  MYC induction increased expression of SRSF1 and PRMT5, but not CLK family kinases. The same samples of Fig 5A were analyzed with RT–PCR.
B  Changes in CPM values of CLK families, PRMT5, and other splicing-related genes calculated from RNA-Seqs data of MYC-inducible SK-MEl-28 cells treated with Dox for 54 h.
C  The numbers of AS events modulated by Dox with a BF > 20 and |ΔPSI| > 0.1 were counted and categorized according to the AS type.
D  Comparison between MYC induction-dependent and CLK inhibitor-dependent alternative splicing events.
E  Effects of T-025 on MYC-inducible SK-MEL-28 cells were analyzed by immunoblotting. SK-MEl-28 cells pretreated with Dox for 48 h were additionally treated with T-025 for 24 h.
F, G  Exon 3 skipping of *NOP16* was induced by T-25. MYC-inducible SK-MEL-28 cells pretreated with Dox were treated with 100 nmol/l T-025 for 6 h. *NOP16* exon 2–3 and 2–4 transcripts were measured by quantitative RT–PCR, and the PSI value of each sample was calculated (F). The copies of exon 2–3 transcripts are shown (G).

Data information: In (A), an unpaired Student's *t*-test was performed. In (F and G), Tukey's test was performed. In (A, F, and G), data are shown as the means ± s.d. of three independent experiments (*n* = 3).
Source data are available online for this figure.

was larger in CLK2 high expressing NCI-H1048 cells, and smaller in CLK2 low expressing 786-O cells. The growth inhibition profile of T-025 in 150 solid cancer cell lines is associated with CLK2 expression. Third, contributions of DYRK1A inhibition for the effects of T-025 were not observed. We revealed that the RNAi-mediated knockdown of DYRK1A did not attenuate the growth inhibitory

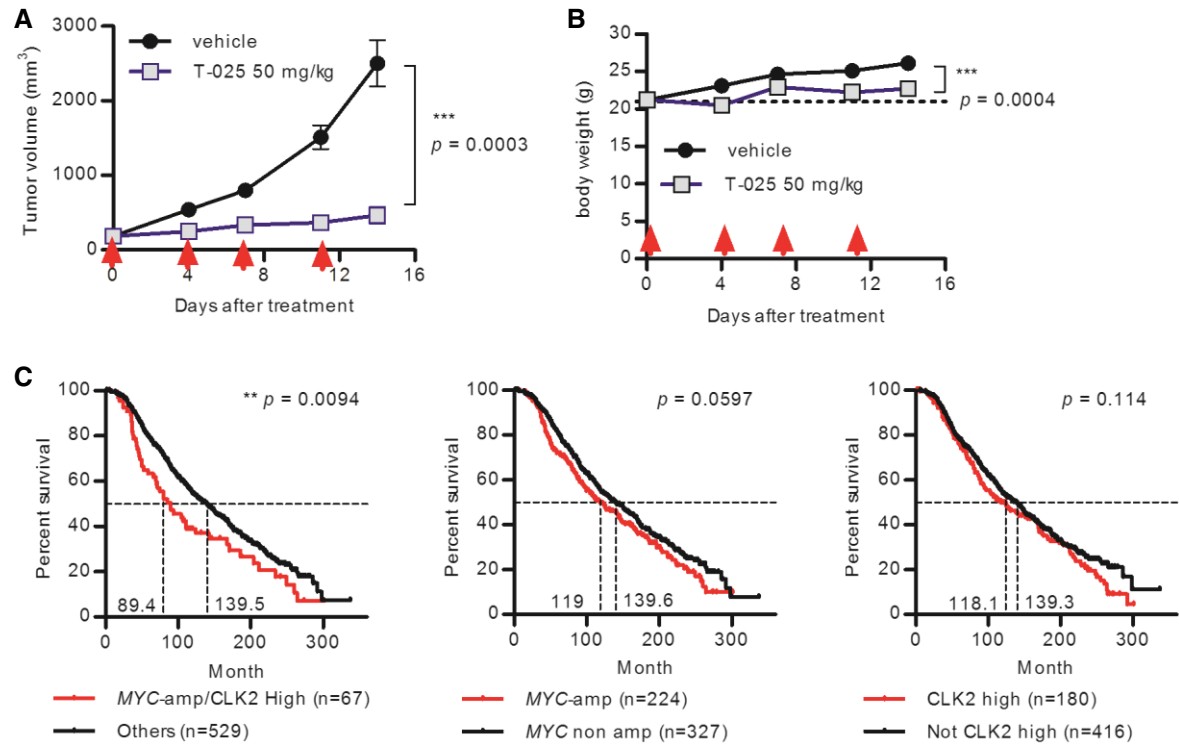

**Figure 7.  T-025 exerted anti-tumor activity in MYC-driven breast cancers.**

A, B   Anti-tumor efficacy of T-025 in a MMTV-*MYC* allograft model. T-025 was administered twice daily on 2 days per week (red arrow). Tumor volume (A) and body weight (B) during the treatment cycle are shown. Data are shown as means ± standard errors of the means (*n* = 8).

C     Kaplan–Meier survival curve of breast cancer patients categorized as an ER-positive, HER2 negative, and high-proliferative subpopulation. Patients were divided into two groups by *MYC*-amplified status and expression level of CLK2. Median survival time of each group was calculated by Prism, and a log rank test was performed.

Data information: In (A and B), an unpaired Student's *t*-test or an unpaired Student's *t*-test with Welch's correction was performed.

effect of T-025. In addition, it has been reported that DYRK1A plays a tumor-suppressive role in AML by downregulating MYC (Liu *et al*, 2014); however, T-025 treatment for 24 h had no effect on the endogenous or exogenous MYC expression in either SK-MEL-28 or U2OS cells. Because the effect of DYRK1A inhibition on cell proliferation is controversial and cellular context dependent (Fernandez-Martinez *et al*, 2015), a broad range of anti-proliferative activity of T-025 also supports our hypothesis that contributions of DYRK1A inhibition on the effects of T-025 are limited and a small molecule inhibitor of CLK has therapeutic potential for cancer.

We show that CLK inhibitor is more effective against MYC-activated cancer, consistent with previous reports that MYC-driven cancers are vulnerable to spliceosome inhibition (Hsu *et al*, 2015; Koh *et al*, 2015). Ectopic activation of MYC enhances sensitivity to CLK inhibitors, as well as 29 *MYC*-amplified solid cancer cell lines are statistically more sensitive to T-025 than other 121 non-*MYC*-amplified solid cancer cell lines. Our result using DYRK1A low-expressed U2OS cells supports our hypothesis that T-025-mediated CLK inhibition causes synergistic cell death with MYC induction. Interestingly, our RNA-Seq data suggested that cellular status of MYC slightly affected the determination of downstream AS events in response to CLK inhibition because T-025 caused similar AS in four tested solid cancer cell lines, including *MYC*-amplified COLO320HSR

cells. The finding that MYC slightly affected CLK inhibitor-dependent AS events was also supported by a previous finding that CLK inhibitor, T3-dependent AS events mostly (~ 75%) overlapped between the hTERT un-transformed fibroblast cells and *MYC*-amplified HCT116 colorectal cancer cells (Funnell *et al*, 2017). We also show that transcriptional regulation of CLK2 and other CLK kinases were independent on MYC activation, degree of T-025-mediated SE was similar regardless MYC induction, and AS events regulated by MYC induction or T-025 rarely overlapped. Based on these results, we hypothesize that synergistic apoptosis induced by MYC induction and CLK inhibition might therefore be because of the concomitant perturbation of different splicing pathways. To support this hypothesis, a synthetic lethality between mutant splicing factors is proposed because of clinical observations that splicing factor mutation exclusively occurs (Yoshida *et al*, 2011; Haferlach *et al*, 2014; Dvinge *et al*, 2016), as well as pharmacological SF3B1-mediated splicing modulators kill cells harboring mutant forms of *SRSF2* or *U2AF1* more effectively (Lee *et al*, 2016; Shirai *et al*, 2017).

Another hypothesis is that CLK2 and MYC cooperatively regulate pre-mRNA biosynthesis and maturation to improve the survival of cancer cells. The SR protein, SRSF1, which is a direct substrate of CLKs and a direct transcriptional target of MYC (Das *et al*, 2012), promotes mammary epithelial cell transformation (Anczukow *et al*,

2012). We showed that the mRNA of *NOP16* is regulated by MYC at the transcriptional level and by CLK inhibitor at the pre-mRNA splicing level. These findings suggest that the CLK family kinases play a critical role in MYC-driven cancers. Importantly, the increased expression of CLK2 worsens the survival of *MYC*-amplified breast cancer patients, while having a slight effect on the survival of non-*MYC*-amplified breast cancer patients. A continuous study to identify the distinct mechanism of cell death mediated by the CLK inhibitor as well as its key downstream targets may demonstrate the mechanism underlying MYC induction-mediated cancer cell vulnerability to CLK inhibition.

One of the limitations of this study is that little investigation was performed regarding the effect on cell survival in response to the CLK inhibitor in hematological cancers. Hematological cancer cell lines may harbor a more powerful factor that affects sensitivity to CLK inhibition. Unbiased bioinformatics analysis also flagged mutations of *CREBBP* or *ROBO2* as sensitive-associated markers (Fig EV1B). Mutations of *CREBBP* are frequently found in acute lymphoblastic leukemia or diffuse large B-cell lymphoma (Mullighan *et al*, 2011; Pasqualucci *et al*, 2011), and mutant-*ROBO2* is detected in patient of MDS (Xu *et al*, 2015). A CLK inhibitor might be more effective against hematological cancers with these mutations. Although we have demonstrated the promising anti-tumor efficacies of T-025 in both a AML cell line xenograft model and a AML PDX model, further studies should be conducted.

Taken together, our findings support the therapeutic potential of CLK inhibitors as a novel splicing modulator. To the best of our knowledge, T-025 is the first CLK inhibitor that exhibits *in vivo* anti-tumor efficacy. Furthermore, our demonstration of a novel synthetic interaction between MYC activation and CLK inhibition provides a better understanding of the functions of the MYC proto-oncogene and CLK, and highlights the specific clinical application of the CLK inhibitor for MYC-activated cancers.

# Materials and Methods

The KINOME*Scan*-based kinase selectivity assay was performed at the DiscoverX Corporation (Fremont, CA, USA). The panel of growth inhibition assay and subsequent bioinformatics analysis (Fig EV1) were performed at Eurofins Inc. (St. Charles, MO, USA), using OncoPanel 240 or 60. Details of the procedures of cell proliferation assay, caspase-3/7 activity assay, immunoblotting, and immunohistochemistry are described in Appendix Supplementary Methods.

## Cell lines

The 293T cell line was purchased from RIKEN (Saitama, Japan). 786-O, BJ, COLO320HSR, MDA-MB-468, MV-4-11, MCF7, NCI-H1048, SK-BR3, SK-MEL-28, U2OS, and Wi38 cells were purchased from the American Type Culture Collection (Manassas, VA, USA). Each line was cultured in the recommended medium, and all were validated as mycoplasma-negative. Other cell lines used in Appendix Fig S3 were described in Appendix Supplementary Methods.

To obtain an MYC-inducible cell line, SK-MEL-28 cells were engineered via stable lentiviral transduction to express an empty pTRIPZ

vector (Open Biosystems/GE Healthcare Dharmacon Inc., Lafayette, CO, USA) or a Tet-inducible promoter-driven *MYC* gene. Infected cells were selected with geneticin for 4 days, left to recover for 24 h, and treated with 0.5 μg/ml Dox (Sigma-Aldrich, St. Louis, MO, USA). U2OS cells were similarly engineered to express an empty pBABE vector (Cell Biolabs, Inc., San Diego, CA, USA) or a MYC-estrogen receptor (MYC-ER) transgene via 3 days of puromycin selection followed by treatment with 0.5 μmol/l tamoxifen (Sigma-Aldrich).

## Antibodies and reagents

The following antibodies were used in this study: anti-phospho-SR monoclonal antibody (Invitrogen/Thermo Fisher Scientific, Carlsbad, CA, USA; 339400, 1/1,000 or 1/5,000 dilution), anti-cleaved PARP1 [Cell Signaling Technology Inc. (CST), Danvers, MA, USA; #9541, 1/2,000 dilution], anti-S6K (CST; #9202, 1/1,000 dilution), anti-SRSF1 (Abcam, Cambridge, UK; ab38017, 1/1,000 dilution), anti-CLK2 (Abcam; ab86147, 1/1,000, 1/2,000 or 1/5,000 dilution), anti-MYC (CST; #5605, 1/1,000 or 1/2,000 dilution), anti-GAPDH (EMD Millipore, Burlington, MA, USA; MB374, 1/2,000 or 1/5,000 dilution), and anti-HSP90 (CST; #4874, 1/5,000 dilution). Horseradish peroxidase-linked anti-rabbit and anti-mouse IgG F(ab)2 fragments (donkey; GE Healthcare, Waukesha, IL, USA) were used as secondary antibodies.

The anti-pS98 CLK2 antibody was used for immunoblotting at 1/1,000 or 1/5,000 dilution and for immunohistochemistry at 1/100 dilution. The detailed procedures for pS98 CLK2 antibody generation and immunohistochemistry are listed in the Appendix Supplementary Methods. Briefly, recombinant CLK2 protein autophosphorylation sites were identified using MS. Tagged-CLK2 phosphorylated sites were also identified from T-025-treated or naïve 293T cells after immunoprecipitation (Appendix Fig S2). Rabbit monoclonal anti-pS98 CLK2 antibodies were generated from splenocytes using Kurosawa's method with modifications (Kurosawa *et al*, 2012).

T-025 and T3 were synthesized at TAKEDA Pharmaceutical Company, Limited. T-025 synthesis is described in the Appendix Supplementary Methods. 4OHT was purchased from Clontech Laboratories (Mountain View, CA, USA).

## Transfection of siRNAs

The transfection of siRNA was performed using the reverse transfection method with Lipofectamine™ RNAiMAX Transfection Reagent (Thermo Fisher Scientific). Cells were transfected with siRNA at 10 nmol/l at 24 h before compound treatment. The following siRNAs purchased from Ambion (Thermo Fisher Scientific) were used: MYC (s9129 and s9130), DYRK1A (s4400 and s4401). Silencer™ Select Negative Control No. 1 siRNA (4390843) and No. 2 siRNA (4390846) were used as non-silencing (NS) siRNAs.

## RNA preparation, quantitative RT–PCR analysis, and calculation of percentage spliced-in value

Total RNA was extracted using the RNeasy Miniprep kit (QIAGEN, Hilden, Germany). cDNAs were synthesized using the TaqMan Reverse Transcription Reagent kit (Applied Biosystems/Thermo

Fisher Scientific). RT–PCR was performed according to Assay-on-Demand, optimized to work with TaqMan Universal PCR MasterMix on a ViiA7 Real-Time PCR system (Applied Biosystems/Thermo Fisher Scientific). The PCR condition was as follows: 2 min at 50°C, 10 min at 95°C, and 40 cycles of 15 s at 94°C and 1 min at 60°C. The FAM-labeled probe and primer mix was used. The amount of FAM fluorescence released from each sample was measured as a function of the PCR cycle number ($C_t$) using the ViiA7 Real-Time PCR system. The variant copy number in each sample was calculated based on the $C_t$ value of each concentration of standard synthesized oligonucleotide. PSI values were calculated as follows:

$$PSI = \text{(copy number of the exon-included variant)}$$
$$/[\text{(copy number of the exon-included variant)}$$
$$+ \text{(copy number of the exon-excluded variant)}]$$

Primers, probes, and standard oligonucleotides for the detection of splicing variants of *RPS6KB1* were as described in a previous report (Araki *et al*, 2015) and those for *BCLAF1* or *NOP16* are described in Appendix Supplementary Methods. TaqMan Gene Expression Assays (Applied Biosystems/Thermo Fisher Scientific) used in this study are also described in Appendix Supplementary Methods.

## Whole, transcriptome RNA sequencing, splicing bioinformatics analysis

The RNA-Seq was performed at Takara Bio Inc. (Shiga, Japan) or Macrogen, Inc. (Seoul, South Korea) by using HiSeq2000 or HiSeq4000 (Illumina Inc., San Diego, CA, USA). Splicing bioinformatics analysis using MISO and pathway analysis using Cytoscape (Shannon *et al*, 2003) were performed as previously described (Funnell *et al*, 2017). Briefly, the BAM alignment files produced by TopHat and the files for AS events (for hg19) provided by MISO were used as inputs. To identify significantly changed AS events between two samples, we used the following criteria: (i) the absolute value of the difference for PSI value (ΔPSI) between two samples was ≥ 0.1; (ii) the sum of inclusion and exclusion reads was > 10, with both inclusion and exclusion reads ≥ 1; and (iii) the Bayes factor (BF), which quantifies the odds of differential regulation occurring, was > 20. For clustering, we used 1,320 splicing events with ΔPSI > 0.4, the BF > 20, and SD of PSI for each splicing event > 0.2. Hierarchical clustering was performed using heatmap3 from the R package.

## Genomic data of cell lines

Genomic data of CCLE, such as CNV and expression, were obtained through cBioPortal for Cancer Genomics (Cerami *et al*, 2012; Gao *et al*, 2013). The definitions of amplification or driver mutations were also obtained from cBioPortal for Cancer Genomics. According to the cBioPortal, putative copy number calls were determined by GISTIC2 using the CCLE segmented data downloaded from the website. Types of mutation were obtained from the OncoKB database (Chakravarty *et al*, 2017). The *MYC* translocation status was categorized using the Guide to Leukemia-Lymphoma Cell Lines (Prof. Dr. Hans G. Drexler).

## CLK2 protein expression of various cell lines

Cell lysates were sampled from 10-cm dish. The culture media and origin of cell lines are described in Appendix Supplementary Methods. We analyzed the CLK2 protein level of 66 cell lines, and the band intensity of each cell line was normalized using band intensities of HCT116 and 786-O in the same gel (Appendix Fig S3). We excluded 11 out of 66 cell lines from the study because of no validation of negative status in the mycoplasma test. The conclusion that CLK2 protein level correlates to the sensitivity to T-025 remained unchanged even when we analyzed using all cell lines.

## Clinical data

Clinical data were also obtained through cBioPortal for Cancer Genomics. The definition of amplification was also obtained from cBioPortal for Cancer Genomics. High CLK2 expression was defined as top 1/3 populations, which was same as the analysis of *in vitro* cell lines. The survival analysis was performed by using GraphPad prism 5.0 (GraphPad Software, Inc., La Jolla, CA, USA).

## *In vivo* experiments

All animal studies were performed in accordance with the protocols approved by the Institutional Animal Care and Use Committees of Takeda Oncology and the Takeda Shonan Research Center. Both facilities are accredited by the Association for the Assessment and Accreditation of Laboratory Animal Care International (AAALAC).

The transgenic MMTV-*MYC* model was generated using the MMTV promoter as previously described (Muller *et al*, 1988) and was obtained from the Mouse Models of Human Cancer Consortium. The MYC-driven breast tumor allograft model was established from a primary tumor of this model (females). Tumor cells were isolated using a cell strainer in the presence of RPMI. Viable cell count was determined, and $0.1 \times 10^6$ cells per mouse were subcutaneously injected in Balb/c nude animals (12-week-old females, Taconic Biosciences, Hudson, NY, USA) for subsequent propagation in the presence of Matrigel (BD, Franklin Lakes, NJ, USA) After two rounds of *in vivo* propagation, the tumor material was cryopreserved in liquid nitrogen. A small piece of the tumor was subcutaneously inoculated into Balb/c nude mice (7- to 8-week-old females), and the tumors were used for efficacy assessment.

To establish the patient-derived xenograft model, a small piece of tumor purchased from Dr. Naoe at Nagoya University was subcutaneously inoculated into NOG mice (7- to 8-week-old females, CIEA Japan, Inc., Kanagawa, Japan). After three rounds of *in vivo* propagation, tumors were used for efficacy assessment.

MDA-MB-468 ($5 \times 10^6$), NCI-H1048 ($5 \times 10^6$), or MV-4-11 ($2 \times 10^6$) cells in Matrigel were subcutaneously inoculated into the left flanks of Balb/c nude mice (7- to 8-week-old females). When xenografted tumors grew to a sufficient volume, the tumor diameter was measured and its volume was calculated as follows:

$$\text{Tumor volume} = \text{long diameter} \times \text{short diameter}$$
$$\times \text{short diameter} \times (1/2)$$

**The paper explained**

**Problem**

Several molecular targeted anti-cancer drugs directly inhibit driver oncogenes such as BRAF, EGFR, or ALK. These drugs show promising efficacy toward cancer patients with an activated oncogene. However, a key treatment strategy for MYC, which is one of the most well-characterized oncogenes, has not been established because of its undruggable function. Further, an inhibitor of splicing regulating kinase CLK2, reported to be oncogenic, has not been developed.

**Results**

We discovered T-025, a novel and potent inhibitor of CLK. Oral treatment with T-025 showed anti-tumor efficacy in the MDA-MB-468 xenograft model, accompanied by the induction of alternative splicing. When T-025 was characterized, CLK2 expression level appeared to be responsible for the sensitivity to T-025. Notably, the CLK inhibitor also exhibited stronger anti-proliferative effects for *MYC*-amplified solid cancer cell lines than for non-*MYC*-amplified solid cancer cell lines. T-025 caused synergistic cell death with MYC activation and showed a promising anti-tumor efficacy in a MYC-driven mouse spontaneous tumor allograft model. Finally, breast cancer patients with both CLK2 high expression and *MYC* amplification showed poor prognosis in the clinical data.

**Impact**

Our study offers the CLK inhibitor as a novel therapeutic option for cancer patients, particularly for poor prognosis patients with *MYC*-amplified and high CLK2-expressing breast cancer.

Mice having a tumor of size approximately 200 mm$^3$ were used in efficacy or *in vivo* pharmacodynamics studies. T-025 was dissolved in 0.5% methylcellulose solution. For the efficacy study, after randomized grouping, mice were treated with T-025 twice daily once or twice weekly for 2 or 3 weeks. The mouse tumor volume and body weight were measured every 2–3 days.

**Statistical method**

Normal distributions were determined by the Kolmogorov–Smirnov test. If normal distribution was determined, then parametric statistical analysis by an unpaired Student's *t*-test, an unpaired Student's *t*-test with Welch's correction, Tukey's test, or Dunnett's test was used based on its variance equality. If normal distribution was not determined, then non-parametric statistical analysis by Mann–Whitney or Steel–Dwass test was used. Statistical tests were performed using GraphPad Prism 5.0 or EXSUS version 8.0 (CAC Croit Corporation, Tokyo, Japan). Details are provided in Appendix Fig S9. n.s., *, **, or *** represent not significant, *P*-value < 0.05, < 0.001, or < 0.0001, respectively.

**Data availability**

The CLK2 structure has been deposited in PDB (Accession code: 5UNP; Appendix Fig S5). The data from the RNA-Seq were deposited in Gene Expression Omnibus (GEO; Accession code: GSE101540, GSE101541).

**Expanded View** for this article is available online.

## Acknowledgements

We would like to thank Dr. Shinsuke Araki, Dr. Karuppiah Kannan, and the members of the Oncology Drug Discovery Unit for providing technical assistance; Dr. Samuel Aparicio at the British Columbia Cancer Agency Vancouver for advice on the manuscript; Dr. Jesse Parry at Eurofins Panlabs for modifying the bioinformatics analysis results; Bi-Ching Sang and Irena Levin for the CLK2-encoding construct; Geza Ambrus-Aikelin for developing the purification method; the staff of the Berkeley Center for Structural Biology for operating the ALS beamline 5.0.3; and Dr. Hiroshi Miyake and Dr. Christopher Claiborne for their guidance and support during the course of this work.

## Author contributions

KI performed the *in vitro* experiments and the bioinformatics analysis, and supervised the biological studies. MY, KN, TT, and DN designed and performed the *in vitro* studies. YY and MT designed and performed the *in vivo* studies. RD, YN, and MK performed the splicing bioinformatics analysis. YK, RM, YI, MA, and HM designed, discovered, and synthesized T-025. KH performed the MS analysis. SM generated the pCLK2 antibody. MI performed the enzymatic assays. MM measured the pharmacokinetics of T-025. MGK, WL, GS, and RT performed CLK2 and T-025 protein purification and crystallography analyses. XH established MYC-inducible cell lines. SP established the MYC-driven breast tumor allograft model. TN supervised the entire study.

## Conflict of interest

All authors are current or previous employees of Takeda Pharmaceutical Company Limited.

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
