## [Review Process File · EMBO Molecular Medicine]

Anti-tumor efficacy of a novel CLK inhibitor via targeting RNA splicing and MYC-dependent vulnerability

Kenichi Iwai, Masahiro Yaguchi, Kazuho Nishimura, Yukiko Yamamoto, Toshiya Tamura, Daisuke Nakata, Ryo Dairiki, Yoichi Kawakita, Ryo Mizojiri, Yoshiteru Ito, Moriteru Asano, Hironobu Maezaki, Yusuke Nakayama, Misato Kaishima, Kozo Hayashi, Mika Teratani, Shuichi Miyakawa, Misa Iwatani, Maki Miyamoto, Michael G. Klein, Wes Lane, Gyorgy Snell, Richard Tjhen, Xingyue He, Sai Pulukuri and Toshiyuki Nomura

Review timeline:

Submission date:	14 October 2017
Editorial Decision:	09 December 2017
Revision received:	19 March 2018
Editorial Decision:	12 April 2018
Revision received:	13 April 2018
Accepted:	18 April 2018

Editors: Roberto Buccione / Céline Carret

Transaction Report:

1st Editorial Decision

09 December 2017

Thank you for the submission of your manuscript to EMBO Molecular Medicine. We have now heard back from the three reviewers whom we asked to evaluate your manuscript.

As you will see, while the reviewers recognize the interest of this manuscript (although reviewer 2 is less convinced overall), they raise many serious concerns.

The three evaluations, albeit with different degrees of emphasis, are remarkably overlapping in terms of the fundamental issues raised. In general, there is agreement that there are many overstated, insufficiently supported conclusions. I will not go into specifics, as the comments are quite detailed and clear. However, I would like to mention a few main points: 1) The specificity of the inhibitor against MYC-driven cancers remains unclear with lack of analysis on the status of MYC and CLK proteins in the cell lines and patient datasets; 2) The molecular connection between MYC and CLK proteins remains unclear; 3) The claim that MYC amplification/mutation predicts sensitivity to the inhibitors is not supported by the data. The reviewers also lament the lack of important controls.

In conclusion, while publication of the paper cannot be considered at this stage, we would be willing to consider a substantially revised submission, with the understanding that the Reviewers' concerns must be addressed in full.

Since the required revision in this case appears to require a significant amount of time, additional work and experimentation, might be technically challenging, and is with no guarantee of success, I would understand if you chose to rather seek publication elsewhere at this stage. Should you do so, and we hope not, we would welcome a message to this effect. Please note that it is EMBO Molecular Medicine policy to allow a single round of revision only and that, therefore, acceptance or rejection of the manuscript will depend on the completeness of your responses included in the next, final version of the manuscript.

As you know, EMBO Molecular Medicine has a "scooping protection" policy, whereby similar findings that are published by others during review or revision are not a criterion for rejection. However, I do ask you to get in touch with us after three months if you have not completed your revision, to update us on the status. Please also contact us as soon as possible if similar work is published elsewhere.

Please note that EMBO Molecular Medicine now requires a complete author checklist (<http://embomolmed.embopress.org/authorguide#editorial3>) to be submitted with all revised manuscripts. Provision of the author checklist is mandatory at revision stage; The checklist is designed to enhance and standardize reporting of key information in research papers and to support reanalysis and repetition of experiments by the community. The list covers key information for figure panels and captions and focuses on statistics, the reporting of reagents, animal models and human subject-derived data, as well as guidance to optimise data accessibility.

We now mandate that all corresponding authors list an ORCID digital identifier. You may do so through our web platform upon submission and the procedure takes < 90 seconds to complete. We also encourage co-authors to supply an ORCID identifier, which will be linked to their name for unambiguous name identification.

Please carefully adhere to our guidelines for authors (<http://embomolmed.embopress.org/authorguide>) to accelerate manuscript processing in case of acceptance.

I look forward to reading your revised manuscript in due time.

***** Reviewer's comments *****

Referee #2 (Remarks):

In this manuscript Iwai et al. have developed a new CLK inhibitor (T-025), elegantly characterized it molecularly and in a series of complementary experiments show interesting evidence of its potential anti-tumor properties. The authors suggest that this compound mechanism of actions goes via the modulation of alternative pre-mRNA splicing functions. These effects are shown in several cellular models, with emphasis in the Myc-driven cancer context. The authors provide in vivo evidence of chemically targeting CLK, and show interesting anti-tumor effects in xenograft and allograft models upon oral administration of the drug. Overall, the manuscript reports convincing evidence of the biological relation between MYC amplification/mutation and CLK inhibition in terms of drug sensitivity, however it is not clear the molecular connection between the two since they are controlling different splicing pathways. Additionally, it would be interesting to analyze whether MYC may transcriptionally regulate CLK as it is described for others putative splicing regulators as PRMT5 (Koh et al. 2015). Therefore, it would be relevant to study the status of MYC and CLK proteins in the different cell lines or patients datasets. Collectively, this manuscript reports an interesting development and a set of relevant findings, however some important points that need further clarification.

Major points

- In the Figure 2A total levels of total CLK2 also decreases when T-025 is added. How come? Is the inhibitor also affecting CLK2 stability? Or as mentioned about, MYC controls CLK2 transcription and there exist a potential positive feedback loop which may explain the interdependency between both although mechanistically their splicing actions are mutually exclusive.
- Authors show that skipping exon 11 of BCLAF1 is one of the most significant effects upon CLK2 inhibition. However, it is not explained the consequence of this aberrant splicing in terms of the functionality of Bcl2 associated transcription factor. Overall, it is unclear why this effects are so specific of transformed cells and do not occur in normal cells. In other words, is the inhibitor altering the spliceosome in non transformed cells (i.e. human mammary epithelial cells or MCF10A immortalized but not transform cells)
- Figure 4 D, E, and F: it is not specified the name of all cancer cell lines used in the study. This is an important issue since the status of MYC in MDA-MB 468 cancer cell line is not showed and it is

relevant the work. If only SKBR3 and AU565 cell lines have a high MYC CVN, and consequently are more sensitive to T-025, why did the authors use MDA-MB 468 cell line to do the experiments of Figure 2 and 3? Please clarify. In addition, confirmation in an alternative breast cancer cell line is needed.

- According to figure 4 D, E, F the enrichment of T-025 sensitivity is related with MYC status in breast cancer cell lines, whereas there is not significant correlation when leukemia or lymphoma cells are analyzed. It is not clear why the authors use melanoma and osteosarcoma cancer cell lines to perform MYC-inducible experiments. Selecting a breast cancer cell line with unaltered MYC and overexpress the protein to test the synergistic effect in combination with T-025 would be a more adequate strategy to fully determine the importance of MYC status for T-025 sensitivity in breast cancer. In addition, why CNV above 8 is needed to explain the effect. 8q amplification in breast cancer (including Myc) is a commonly reported event, however none of the cell lines reported to have such gain seem to respond. Please clarify.

- MMTV- MYC transgenic mice tumors are used to fully validate the CLK inhibitor in vivo by using allograft models. However, the authors should explain why they used this system instead treating MMTV-MYC mice with T-025, which is more physiologically reliable. Please clarify

- It is unclear what is the role of DYRK1A in all the effects reported. Given the kinetic properties it is almost as good as a target as CLK2 and although it has been previously reported to act as a tumor suppressor, no experiments role out its contribution. Does the T-025 inhibitory effects persist in the absence of DYRK1A by means of knockdown?

- It is unclear what is the contextual clinical relevance of the current findings in terms of patient population and potential patient stratification to be treated based on MYC status.

Minor points

- Figure 2 D and supplementary S2C: control of cells treated with DMSO should be used in parallel.
- Line 111: "largely" is repeated in the same sentence and seems inappropriate to describe an event which either occur or does not.

- Figure 2C is not cited in the manuscript

- Line 126: Figure 2D is referred but there is no relation with the text in this part of the manuscript.

- Statistics in figure 6 B, C, D, and F is missing.

- In the discussion, please avoid interpreting the magnitude of your data relevance (line 229-232). In addition please rephrase the text between lines 236 and 239. It is unclear what is the message to be conveyed.

Referee #3 (Remarks):

Iwai and colleagues synthesized a CLK inhibitor (T-025) capable of modulating CLK-mediated splicing. Authors claim that T-025 exhibits higher anti-proliferative activity specifically against MYC-driven cancers, given the previously known susceptibility of these tumors to spliceosome inhibition. The concept of targeting a vulnerability of MYC is pertinent given the relevance of this transcription factor in cancer, and T-025 does seem to have some selective inhibitory activity against CLK-family members. Nevertheless the experiments presented in this manuscript do not fully demonstrate the specificity of this inhibitor against MYC-driven cancers. Overall the manuscript is poorly discussed, has some statements not supported by the experimental data, and lacks flow (several figure panels are not called chronologically in the text).

It is puzzling why the focus of the work is on MYC when (1) authors find several other biomarker candidates associated with T-025 sensitivity besides MYC mutation/amplification (Figure 4C); and (2) the correlation between MYC amplification and T-025 sensitivity is only statistically significant in breast cancer (with only 2 cell lines with MYC amplification), and not significant in the other two tested tumor types: leukemia or lymphoma. It is also unclear why did authors chose these three tumor types to perform this analysis, and should be explained. These results do not give strength to the hypothesis and indicate that the correlation sensitivity to T-025 and MYC levels is at best context dependent.

Authors claim that that CLK inhibition functions as a novel pre-mRNA splicing modulation-based anti-cancer strategy for MYC-driven cancers. It is not clear why did authors chose these cell lines for the experiments in Figures 2, 3 and 6. Are they MYC-driven cancer cells? How do these models support the hypothesis? Did authors performed experiments with T-025 in non-MYC-driven cancers? Authors miss a clear loss of function experiment in order to conclude the specificity of T-025 in MYC-driven cancers.

What was the criteria used to terminate the in vivo experiments, given that their duration and, as a consequence, their outcome are so variable. For example, in Figure 3D the experiment lasted 22 days with control tumors rather small (volume <300 mm³) and a substantial tumor growth impairment. However the experiment of Figure S3C was kept for approximately 55 days (volume of control tumors >2000 mm³) and despite the initial response, tumors from the treated arms seem to have developed resistance to T-025. This observation suggests that T-025 single treatment will not treat these tumors. Is the experiment represented in Figure S3C illustrating what would have happened with the other tumor models if these would have been kept for longer time in experiment? These discrepancies should at least be discussed in the manuscript.

Authors start the Discussion section by writing that "Our findings clearly demonstrate that CLK inhibition is a highly effective treatment strategy against MYC-activated cancers". With the above mentioned, this is clearly an overstatement.

Figure 6E - What is the rationale behind a gain of function experiment, when T-025 sensibility does not seem to correlate with MYC levels demonstrated in Figure 4?

Figure 2B - how long was the T-025 treatment for this IC₅₀ calculation? Also 72h, as in the 240-cell panel experiment?

Figure 3B - scale bar is missing

Figure S3B and S3C - how many mice were used per cohort?

Figure S3D is missing.

Referee #4 (Remarks):

The authors describe a novel inhibitor of the CLK2 kinase, an oncogenic kinase which regulates RNA splicing. They explore the idea that these compounds may be effective against MYC-driven tumors on the basis that MYC may generally deregulate basal transcription and predispose tumors cells to splicing disturbance. The concept to target MYC-driven tumors via interfering with alternate splicing has been suggested previously, but still remains interesting and some of the data presented here are indeed very interesting. In the current form, the manuscript raises a number of major issues that need to be addressed prior to publication. For example, the major claim that MYC amplification or mutation predicts sensitivity to the inhibitors is not supported by the panels shown in Figure 4. Also, many elements and panels are very hard to understand.

Major issues

Figure 1 states that all other kinases are inhibited to less than 90% at 300nM, but this still means that many kinases are massively inhibited at the concentrations used in the subsequent experiments. In my view, the authors need to provide clear data that the observed data are on-target activities of the inhibitors. For example, the crystal structure should enable the authors to identify a point mutant allele of CLK2 that is more resistant to the inhibitor. Failing this, the authors should show that depletion/deletion of CLK2 has similar effects on splicing and that the inhibitor has less/no additional effects in CLK2 depleted/deleted cells.

Figure 2: Figure 2B does not match Figure 2A.

BF is undefined and Δ PSI is not clearly defined. If Δ PSI is really "percentage splice-in", none of the effects is significant. The normalization of Δ PSI is unclear.

In Figure 2E, the difference between 100nM and 300nM is unclear.

It would be important to see immunoblots of some of the affected proteins to judge the significance and magnitude of the effects.

Figure 4:

The resolution of panels A and - in particular B- is too low, looks like these are screenshots. I cannot read the legend of panel B.

Which of the circles- the size of which is unexplained - is the MYC one is not clear. Neither seems to match the values given in panel C. Also, the rationale why two test are used, is unclear.

Panels D,E,F give p-values for the mutation or amplification status, which makes no sense. The authors should just determine what it is and give the numbers and mutations. Critically, none of these three panels supports the claim that MYC-amplified or mutated cancer cells are particularly sensitive to the inhibitor. For this, much stronger data would be necessary.

In the description of this figure, the authors state: "Both MYC amplification and mutation lead to MYC activation": I do not think that this is consistent with the literature.

The analysis suggests that several mutations can enhance sensitivity to these drugs, some with significantly lower p-values than MYC. It would be good to see one or two of the other predictions tested.

Figure 6

Why is the colour code different from panel 2A?

As before, it would be important to see immunoblots of some of the affected proteins to judge the significance and magnitude of the effects.

The authors should also perform a bioinformatics analysis to provide some insight which exons are skipped? Do they have a particular feature? Are they in genes of a specific functional category?

How does exon skipping in these genes affect cell growth?

Immunoblots of MYC levels relative to endogenous MYC levels of some tumor cells should be shown to judge the magnitude of overexpression.

1st Revision - authors' response

19 March 2018

In light of the referees' comments, we took another close look at our data, including new experimental data and the analysis results thereof. From this, we found that the higher growth-suppressive effect of T-025 in MYC-activated cancer cell lines was not observed in breast cancer cells, but might nevertheless be applicable to other solid cancers. We have revised the manuscript substantially, particularly the bioinformatics analysis, the explanation of the connection between sensitivity to T-025 and the expression of CLK2, and details of the relationship between MYC and CLKs as follows:

- T-025 was more effective in all MYC-amplified solid cancer cell lines, not just in breast cancer cell lines (Fig. 3B).
- There was a statistically significant correlation between growth-inhibitory sensitivity to T-025 and CLK2 mRNA expression (Fig. 3C) or CLK2 protein level (Fig. 4A). In addition, the degree of alternative splicing (AS) in response to T-025 depended on the CLK2 protein level in each cell (Fig. 4).
- We have also added results regarding the molecular relationship between MYC and CLKs (Fig. 6), and clinical observations based on MYC amplification and CLK2 expression (Fig. 7C).

Revised manuscript Fig No.	Initial manuscript Fig No.	Main data and Changes
Figure 1	Figure 1	Structure of T-025 (Not changed)
Figure 2	Figure 2 and Figure 3	The results of T-025 in MDA-MB-468 cells both in vitro and in vivo are shown.
Figure 3 (EV1, EV2, and EV5)	Figure 4	We have added new analysis results regarding Oncopanel 240 and modified figures (e.g. analysis of CLK2 expression, the name of cell lines)
Figure 4	Figure S2 and new	We have added new results regarding CLK2

(and EV3) Figure 5	data Figure 6 and new data	expression and AS sensitivity to T-025 MYC activation and sensitivity to T-025 We have added results obtained using siRNAs
Figure EV4	Figure 5 and new data	Comparison to another CLK inhibitor We have added new data regarding the comparison of AS
Figure 6 Figure 7	New data Figure 7 and new data	Molecular connection between CLKs and MYC We have added results of our analysis of the clinical database

In response to the points raised by the referees, we would like to make the following remarks.

A) The bioinformatics analysis showed that *MYC*-amp/mut is associated with sensitivity to T-025

We apologize for the unclear figures and results. In the original manuscript, we used a figure from Eurofins Inc. to illustrate the bioinformatics analysis. The reason we chose breast cancer, leukemia, and lymphoma cell lines is that the literature showed that breast and hematological cancers with *MYC* activation are vulnerable to spliceosome inhibition (Hsu et al, 2015; Koh et al, 2015).

In this revised manuscript, to make the results and interpretation clearer, we have re-analyzed the data using 169 (150 solid and 19 hematological) of the total 240 cell lines whose genomic data could be obtained from the CCLE database (Fig EV2A). The genomic data as well as the definitions of amplification and driver mutation were obtained from the cBioportal website (<http://www.cbioportal.org/index.do>). In the cBioportal, to determine copy number status, GISTIC2 was run using the GenePattern website, using CCLE segmented data downloaded from the CCLE website. The mutations were cross-checked with OncoKB (<http://oncokb.org/#/>) to determine driver mutations. We also checked *MYC* translocation in the Guide to Leukemia-Lymphoma Cell Lines by Dr. Hans G. Drexler (2nd edition).

No solid cancer cell lines with an *MYC*-driver mutation were observed in the 150 cell lines. Importantly, solid cancer cell lines with amplified *MYC* ($n = 29$) were significantly more sensitive to T-025 than solid cancer cell lines without amplified *MYC* ($n = 121$) (** $p = 0.0042$) (Fig. 3B). On the contrary, we did not observe higher sensitivity to T-025 in *MYC*-altered hematological cancer cell lines (amplified, driver-mutated, or translocated *MYC*) (Fig. EV2B). Based on these analytical results, we concluded that various solid cancer cell lines, not only breast cancer cell lines, with *MYC* amplification were more sensitive to T-025 than solid cancer cell lines without *MYC* amplification.

B) The reasons we chose the cell lines

MDA-MB-468 cells

We apologize for our unclear explanations, particularly regarding the purpose of each experiment. We performed our fundamental CLK research using MDA-MB-468 cells (Araki et al, 2015) and found that other CLK inhibitors induced dephosphorylation of SR proteins, global alternative splicing, and cell death in MDA-MB-468 cells. In addition, RNAi-mediated knockdown of CLK1 and/or CLK2 causes exon 7 of *RPS6KB1* to be skipped in MDA-MB-468 cells. Although MDA-MB-468 cells do not have amplified *MYC*, we used this cell line to see whether treatment with T-025 resulted in the CLK inhibition-mediated phenotypes that had already been reported.

We have revised the manuscript to improve clarity of the background information and facilitate readers' understanding.

MYC-inducible cell lines of non-breast cancers

An unbiased analysis performed by Eurofins, Inc. showed that various cancer cell lines with *MYC*-amplifications/mutations were more sensitive to T-025 than those without *MYC*-alteration. Because *MYC*-inducible SK-MEL-28 cells (He et al, 2017) and U2OS cells had already been established when we obtained the Oncopanel result, we used this cell line to confirm the biological relationship between *MYC* and CLK inhibition. In addition, U2OS cells were used to detect the effect of DYRK1A.

Although our new analysis revealed that not only breast cancer cell lines but also solid cancer cell lines with *MYC* amplification were more sensitive to T-025 than solid cancer cell lines without *MYC* amplification, we performed an siRNA experiment using *MYC*-amplified breast cancer cell lines (Fig 5I).

Referee #2 (Remarks):

In this manuscript Iwai et al. have developed a new CLK inhibitor (T-025), elegantly characterized it molecularly and in a series of complementary experiments show interesting evidence of its potential anti-tumor properties. The authors suggest that this compound mechanism of actions goes via the modulation of alternative pre-mRNA splicing functions. These effects are shown in several cellular models, with emphasis in the Myc-driven cancer context. The authors provide in vivo evidence of chemically targeting CLK, and show interesting anti-tumor effects in xenograft and allograft models upon oral administration of the drug. Overall, the manuscript reports convincing evidence of the biological relation between *MYC* amplification/mutation and CLK inhibition in terms of drug sensitivity, however it is not clear the molecular connection between the two since they are controlling different splicing pathways. Additionally, it would be interesting to analyze whether *MYC* may transcriptionally regulate CLK as it is described for others putative splicing regulators as *PRMT5* (Koh et al. 2015). Therefore, it would be relevant to study the status of *MYC* and CLK proteins in the different cell lines or patients datasets. Collectively, this manuscript reports an interesting development and a set of relevant findings, however some important points that need further clarification.

Major points

- In the Figure 2A total levels of total CLK2 also decreases when T-025 is added. How come? Is the inhibitor also affecting CLK2 stability? Or as mentioned about, *MYC* controls CLK2 transcription and there exist a potential positive feedback loop which may explain the interdependency between both although mechanistically their splicing actions are mutually exclusive.

We thank the referee for these comments. It has already been reported that phosphorylation of CLK2 at pS98 contributes to the protein stability of CLK2, and CLK inhibitor (TG003) promotes degradation of the CLK2 protein (Rodgers et al, 2010). Our results are consistent with this report. CLK2 FPKM values calculated from RNA-Seq (Fig 2C) showed that the expression level of CLK2 was not decreased in response to T-025 treatment (below).

The decreased level of CLK2 protein in response to T-025 is thought to be due to reduced protein stability as a result of inhibition of kinase activity, rather than suppression of CLK2 transcription. We have explained this fully in the revised manuscript.

In light of your comments, we tested whether *MYC* activation increased CLK2 mRNA. However, we found no upregulation of CLK2 following *MYC* induction, although we confirmed upregulation of *PRMT5* in our cell line, indicating that CLK2 is not a direct transcriptional target of *MYC* (Fig. 6A). In addition, we did not observe stabilization of the CLK2 protein after *MYC* activation (Appendix Fig. 5B and Fig 6E).

- Authors show that skipping exon 11 of *BCLAF1* is one of the most significant effects upon CLK2 inhibition. However, it is not explained the consequence of this aberrant splicing in terms of the functionality of Bcl2 associated transcription factor. Overall, it is unclear why this effects are so specific of transformed cells and do not occur in normal cells. In other words, is the inhibitor

altering the spliciosome in non transformed cells (i.e. human mammary epithelial cells or MCF10A immortalized but not transform cells)

Although alternative splicing of *BCLAF1* is reported to have an additional function in colorectal cancer (Zhou et al, 2014), the reported AS variant was different from the AS variant (lacking exon 11) produced by T-025. We did not investigate the implications of the *BCLAF1* variant lacking exon 11; however, in general, our previous reports show that aberrant AS variants caused by CLK inhibitors tend to lose their stability and degrade rapidly (Araki et al, 2015).

In terms of the difference between cancer cells and normal cells, CLK inhibitor T3 causes similar AS in both cancer cell lines (HCT116) and untransformed cell lines (hTERT) (Funnell et al, 2017). Our results in the present study also showed that the CLK inhibitor causes similar AS in all tested cell lines. These data suggest that downstream AS in response to CLK inhibition is common in normal and cancer cells.

Unfortunately, we do not have a clear answer to your question about the effect of T-025 being so specific to transformed cells. However, we did show that untransformed normal cell lines (Wi38 and BJ) had relatively low expression of CLK2 and less growth-suppressive sensitivity to T-025 (Fig 4B). Our data showed that the expression level of CLK2 in the cancer cell line was correlated with the degree of AS in response to T-025 (Fig 4E), suggesting that increased expression of CLK2 rendered cancer cells more sensitive to T-025. In addition, we revealed that MYC activation and the CLK inhibitor synergistically caused cell death (Fig. 5). Based on these data, we hypothesize that short-term treatment with T-025 causes apoptosis only in cancer cells with high expression of CLK2 or with activated MYC, but not in normal tissue. Therefore, in the mouse model, intermittent treatment with T-025 exhibited *in vivo* anti-tumor properties without severe body weight loss.

- Figure 4 D, E, and F: it is not specified the name of all cancer cell lines used in the study. This is an important issue since the status of MYC in MDA-MB 468 cancer cell line is not showed and it is relevant the work. If only SKBR3 and AU565 cell lines have a high MYC CVN, and consequently are more sensitive to T-025, why did the authors use MDA-MB 468 cell line to do the experiments of Figure 2 and 3? Please clarify. In addition, confirmation in an alternative breast cancer cell line is needed.

As we mentioned above, although MDA-MB-468 cells do not have amplified *MYC*, we used MDA-MB-468 cells to check the CLK inhibition-dependent phenotype, which had already been reported using different CLK inhibitors or an RNAi-mediated CLK knockdown. The *in vitro* growth-suppressive sensitivity of SKBR3 or AU565 to T-025 was equivalent to that of MDA-MB-468. The name of the cell line is indicated in Fig. EV5.

To demonstrate the effect of T-025 in another MYC-driven breast cancer, we conducted *in vivo* anti-tumor efficacy studies in SUM-159 xenografts. The SUM-159 cell line has been reported as an MYC-driven breast cancer cell line (Hsu et al, 2015), and was not included among the 240 tested cancer cell lines. As shown below, T-025 showed significant anti-tumor properties in this model, suggesting that T-025 is effective for MYC-driven breast cancer cell line xenografts.

- According to figure 4 D, E, F the enrichment of T-025 sensitivity is related with MYC status in breast cancer cell lines, whereas there is not significant correlation when leukemia or lymphoma

cells are analyzed. It is not clear why the authors use melanoma and osteosarcoma cancer cell lines to perform MYC-inducible experiments. Selecting a breast cancer cell line with unaltered MYC and overexpress the protein to test the synergistic effect in combination with T-025 would be a more adequate strategy to fully determine the importance of MYC status for T-025 sensitivity in breast cancer. In addition, why CNV above 8 is needed to explain the effect. 8q amplification in breast cancer (including Myc) is a commonly reported event, however none of the cell lines reported to have such gain seem to respond. Please clarify.

The reasons for our choice of non-breast MYC-inducible cancer cell lines are detailed above. Regarding the CNV, we re-analyzed the Oncopanel data with a new definition of amplification based on a commonly used method. We found that *MYC*-amplified solid cancer cell lines were more sensitive to T-025 than non-*MYC*-amplified solid cancer cell lines. Among breast cancer cell lines, *MYC*-amplified cell lines tended to be sensitive to T-025, but not significantly ($p = 0.0712$, Fig. EV5).

- MMTV- MYC transgenic mice tumors are used to fully validate the CLK inhibitor in vivo by using allograft models. However, the authors should explain why they used this system instead treating MMTV-MYC mice with T-025, which is more physiologically reliable. Please clarify

We agree with this reviewer's comment. Using MMTV-MYC mice to evaluate T-025 is physiologically more reliable than using an allograft model. However, it is relatively easy to handle the experiment using the allograft, and this enabled us to obtain robust results and to conduct a multiple-arm study to evaluate multiple compounds. Therefore, we established the allograft model before this CLK-MYC study. When we initiated this study, we did not maintain the live mice; therefore, we performed the experiment using frozen allograft tumors.

- It is unclear what is the role of DYRK1A in all the effects reported. Given the kinetic properties it is almost as good as a target as CLK2 and although it has been previously reported to act as a tumor suppressor, no experiments rule out its contribution. Does the T-025 inhibitory effects persist in the absence of DYRK1A by means of knockdown?

We conducted a growth inhibition assay of T-025 in MCF7 or SKBR3 cells pre-transfected with DYRK1A siRNA. As shown in Fig 5I and Appendix Fig S4B, knockdown of DYRK1A did not attenuate the growth-inhibitory effect of T-025. In addition, our data suggested that the effect of T-025 was mainly attributable to CLK inhibition for the following reasons;

- Growth inhibition sensitivity of T-025 was correlated with the CLK2 expression level, not to that of other CLK or DYRK1 family kinases (Fig. EV1B).
- The degree of AS in response to T-025 was correlated with the expression level of CLK2 in each cell line (Fig. 4).
- U2OS cells are reported to be one of the cell lines with the lowest expression and CNV of DYRK1A (Appendix Fig S4A). Our data indicated that U2OS cells treated with T-025 showed growth suppression and MYC-dependent increased sensitivity to T-025 (Fig 5).

These data suggest that the effect of T-025 was mainly attributable to CLK inhibition and unlikely to be due to inhibition of DYRK1A. We added these thoughts about the effect of DYRK1A inhibition to the discussion part of our revised manuscript.

- It is unclear what is the contextual clinical relevance of the current findings in terms of patient population and potential patient stratification to be treated based on MYC status.

In the breast cancer patient database, *MYC* amplification was observed in approximately 20% of patients. When we divided these patients based on their CLK2 mRNA levels, patients with both high CLK2 and *MYC* amplification had the worst prognoses (Fig. 7C, Appendix Fig S6B). We would expect the CLK inhibitor to be more effective in these patients. Unfortunately, we could not find a clinical database that held the CLK2 protein expression data, although we showed that the protein level of CLK2 was also correlated to the sensitivity to T-025 in this revised manuscript.

Minor points

- Figure 2 D and supplementary S2C: control of cells treated with DMSO should be used in parallel.

We apologize that we did not explain normalization clearly enough. These figures show the number of AS events in which the PSI difference (Δ PSI) between DMSO and the treatment sample (e.g. 30 nmol/L) was larger than 0.1. These figures do not indicate the PSI values at each concentration. The PSI and Δ PSI values of *BCLAF1* exon 11, as an example, have been shown below.

We have modified the figures (Figs 2C, 4C, 6C) to make this clear.

- Line 111: "largely" is repeated in the same sentence and seems inappropriate to describe an event which either occur or does not.
- Figure 2C is not cited in the manuscript
- Line 126: Figure 2D is referred but there is no relation with the text in this part of the manuscript.
- Statistics in figure 6 B, C, D, and F is missing.
- In the discussion, please avoid interpreting the magnitude of your data relevance (line 229-232). In addition please rephrase the text between lines 236 and 239. It is unclear what is the message to be conveyed.

We thank the referee for these comments. We have corrected the manuscript accordingly. Statistical tests in Fig 5C, G, and I are shown in Appendix Fig S as IC_{50} values and 95% confidential intervals.

Referee #3 (Remarks):

Iwai and colleagues synthesized a CLK inhibitor (T-025) capable of modulating CLK-mediated splicing. Authors claim that T-025 exhibits higher anti-proliferative activity specifically against MYC-driven cancers, given the previously known susceptibility of these tumors to spliceosome inhibition. The concept of targeting a vulnerability of MYC is pertinent given the relevance of this transcription factor in cancer, and T-025 does seem to have some selective inhibitory activity against CLK-family members. Nevertheless the experiments presented in this manuscript do not fully demonstrate the specificity of this inhibitor against MYC-driven cancers. Overall the manuscript is poorly discussed, has some statements not supported by the experimental data, and lacks flow (several figure panels are not called chronologically in the text).

It is puzzling why the focus of the work is on MYC when (1) authors find several other biomarker candidates associated with T-025 sensitivity besides MYC mutation/amplification (Figure 4C); and (2) the correlation between MYC amplification and T-025 sensitivity is only statistically significant in breast cancer (with only 2 cell lines with MYC amplification), and not significant in the other two tested tumor types: leukemia or lymphoma. It is also unclear why did authors chose these three tumor types to perform this analysis, and should be explained. These results do not give strength to the hypothesis and indicate that the correlation sensitivity to T-025 and MYC levels is at best context dependent.

We apologize for any unclear explanations, particularly regarding the purpose of each experiment.

(1) We set up a large-scale growth inhibition panel and found several biomarker candidates just before the *Nature* papers were published. Since the external publications reflected the internal data, we focused on *MYC* for further validation.

(2) As we have detailed above, we re-analyzed the data using another definition of *MYC* amplification. We found solid cancer cell lines with *MYC* amplification were more sensitive to T-025 than other solid cancer cell lines, but not hematological cancer cell lines with *MYC* alterations (Fig 3B and Fig EV2).

Authors claim that that CLK inhibition functions as a novel pre-mRNA splicing modulation-based anti-cancer strategy for *MYC*-driven cancers. It is not clear why did authors chose these cell lines for the experiments in Figures 2, 3 and 6. Are they *MYC*-driven cancer cells? How do these models support the hypothesis? Did authors performed experiments with T-025 in non-*MYC*-driven cancers? Authors miss a clear loss of function experiment in order to conclude the specificity of T-025 in *MYC*-driven cancers.

As mentioned above, we performed the experiments using MDA-MB-468 to show the CLK inhibition-mediated phenotypes of T-025 in comparison with those of RNAi, which have already been published. These data were not intended to support the *MYC* hypothesis.

As *MYC*-inducible cell lines, as detailed above, we used non-*MYC*-driven cancer cell lines to determine the effect of *MYC* activation.

We also performed the *MYC* knockdown experiment in the *MYC*-amplified breast cancer cells SKBR3 or MCF7, and found that reduced expression of *MYC* rendered SKBR3 or MCF7 cells less sensitive to T-025 (Fig. 5I and Appendix Fig S4B).

What was the criteria used to terminate the in vivo experiments, given that their duration and, as a consequence, their outcome are so variable. For example, in Figure 3D the experiment lasted 22 days with control tumors rather small (volume <300 mm³) and a substantial tumor growth impairment. However the experiment of Figure S3C was kept for approximately 55 days (volume of control tumors >2000 mm³) and despite the initial response, tumors from the treated arms seem to have developed resistance to T-025. This observation suggests that T-025 single treatment will not treat these tumors. Is the experiment represented in Figure S3C illustrating what would have happened with the other tumor models if these would have been kept for longer time in experiment? These discrepancies should at least be discussed in the manuscript.

We apologize for the unclear statement in the manuscript. However, the referee's comment that the MV-4-11 tumors became resistant is not true. Drug treatment was performed during the first two weeks only, and after that, we merely monitored tumor sizes and body weights without treatment to observe regrowth of the tumors, because the MV-4-11 tumors were very thin and had almost disappeared by day 15.

We generally studied anti-tumor efficacy for two or three weeks. Since the growth of MDA-MB-468 tumors was slow, we continued to treat them with T-025 for a total of three weeks. We never treated mice with T-025 for more than three weeks.

We have modified the figures to show efficacy during the treatment cycles to improve readers' understanding.

Authors start the Discussion section by writing that "Our findings clearly demonstrate that CLK inhibition is a highly effective treatment strategy against *MYC*-activated cancers". With the above mentioned, this is clearly an overstatement.

We have corrected the manuscript according to your comment.

Figure 6E - What is the rationale behind a gain of function experiment, when T-025 sensibility does not seem to correlate with *MYC* levels demonstrated in Figure 4?

As mentioned above, because the external publications (Hsu et al, 2015; Koh et al, 2015) reflected the internal data, we performed the gain of function experiments. We did not have a rationale to perform the *MYC* experiments if external publications had not been published. We confirmed that *MYC* induction in our SK-MEL28 cells significantly increased total RNA synthesis using the Click-iT™ RNA kit (see below).

Therefore, our data supported the hypothesis that MYC activation-mediated RNA deregulation, such as increased global RNA synthesis (RNA amplification) and modulation of AS renders cancer cells vulnerable to spliceosome inhibition.

Figure 2B - how long was the T-025 treatment for this IC50 calculation? Also 72h, as in the 240-cell panel experiment?

Yes, 72 h.

Figure 3B - scale bar is missing

Figure S3B and S3C - how many mice were used per cohort?

Figure S3D is missing.

We thank the referee for these comments. We have corrected the manuscript accordingly.

Referee #4 (Remarks):

The authors describe a novel inhibitor of the CLK2 kinase, an oncogenic kinase which regulates RNA splicing. They explore the idea that these compounds may be effective against MYC-driven tumors on the basis that MYC may generally deregulate basal transcription and predispose tumors cells to splicing disturbance. The concept to target MYC-driven tumors via interfering with alternate splicing has been suggested previously, but still remains interesting and some of the data presented here are indeed very interesting. In the current form, the manuscript raises a number of major issues that need to be addressed prior to publication. For example, the major claim that MYC amplification or mutation predicts sensitivity to the inhibitors is not supported by the panels shown in Figure 4. Also, many elements and panels are very hard to understand.

Major issues

Figure 1 states that all other kinases are inhibited to less than 90% at 300nM, but this still means that many kinases are massively inhibited at the concentrations used in the subsequent experiments. In my view, the authors need to provide clear data that the observed data are on-target activities of the inhibitors. For example, the crystal structure should enable the authors to identify a point mutant allele of CLK2 that is more resistant to the inhibitor. Failing this, the authors should show that depletion/deletion of CLK2 has similar effects on splicing and that the inhibitor has less/no additional effects in CLK2 depleted/deleted cells.

Regarding the suggested experiments, we found that ectopic expression of wild-type CLK2 or RNAi-mediated reduction of CLK2 impaired the growth of cells. Therefore, we believe it would be difficult to judge the effect of T-025 in these growth-impaired cells.

Regarding the similarity between T-025 and CLK2 deletion, we showed that the skipped exon 7 of *PRS6KBI* was observed by RNAi of CLK2 and T-025. In addition, we found that T-025 and another CLK inhibitor T3 showed similar AS (Fig EV4C). It had already been reported that ASs induced by T3 largely overlapped those induced by CLK1/2/3/4 RNAi (Funnell et al, 2017); hence, we deduced that T-025 and depletion of CLKs caused a similar splicing effect.

To add support to the idea that this effect was caused by CLK inhibition, we noted that the spontaneous expression level of CLK2 is correlated to the growth-suppressive sensitivity and the degree of AS in response to T-025 treatment (Figs 3C and 4A). Furthermore, T3, another CLK inhibitor, exhibited a similar profile of growth suppression of 60 cancer cell lines, modulation of AS, and increased MYC-dependent vulnerability (Fig EV4).

In addition, HIPK2 is one of the most potent kinases inhibited by T-025, apart from CLK and DYRK family kinases. However, we did not observe suppression of Ser46 due to p53 (pS46 p53) phosphorylation, at a direct phosphorylation site of HIPK2, up to 1000 nmol/L (see below, from Fig. 2A). Since pS46 p53 was also reported to be phosphorylated by other kinases, this result did not clearly show selectivity, but it could support the observation that T-025 did not inhibit HIPK2 in cells up to 1000 nmol/L.

These data suggest that the effect of T-025 was mainly attributable to CLK inhibition.

Figure 2: Figure 2B does not match Figure 2A.

In quantifying the bands of pCLK2, the background of this immunoblotting around 50 kDa was relatively high and the band was shifted in the T-025 treated samples (below); thus, we could not remove the background effect, because the relative band intensity did not reduce to near 0.

BF is undefined and Δ PSI is not clearly defined. If Δ PSI is really "percentage splice-in", none of the effects is significant. The normalization of Δ PSI is unclear. In Figure 2E, the difference between 100nM and 300nM is unclear.

According to the MISO software, the Bayer Factor (BF) represents the weight of the evidence in the data in favor of differential expression versus the weight of evidence not in favor. For example, a

Bayes factor of 2 would mean that the isoform/exon is two times more likely to be differentially expressed than not. We have added comments about this in the materials and methods section of the revised manuscript.

We have also added PSI values to the sashimi plot of BCLAF1 to show the difference between 100 nmol/L and 300 nmol/L (Appendix Fig S1E).

As we have mentioned above (in answer to referee #2), these figures show the number of AS events whose PSI difference between the DMSO and treatment sample (e.g. 30 nmol/L) is larger than 0.1 ($\Delta\text{PSI} > 0.1$). We have modified the figures to clarify this.

$\Delta\text{PSI} > 0.1$ is a widely used cutoff value when comparing the splicing pattern in SRSF2 P95H cells and in wild-type cells (Kim et al, 2015) or between the vehicle treatment sample and the E7107 (SF3B1-mediated splicing modulator) treatment sample (Lee et al, 2016).

It would be important to see immunoblots of some of the affected proteins to judge the significance and magnitude of the effects.

We have added the data from the immunoblot to Appendix Fig. S1D. The protein levels of S6K, of which exon 7 was skipped by T-025, were reduced by T-025 treatment. We also performed immunoblotting of MYC-inducible cells treated with T-025 and presented the data in Fig 6E.

Figure 4:

The resolution of panels A and - in particular B- is too low, looks like these are screenshots. I cannot read the legend of panel B.

Which of the circles- the size of which is unexplained - is the MYC one is not clear. Neither seems to match the values given in panel C. Also, the rationale why two test are used, is unclear.

We apologize for the unclear figure. The unbiased analysis and the figure were performed at Eurofins Inc. The exact reason why they used two tests are used is not clear, but statistical evaluation by using two different tests, a student's t-test (parametric test) and a fisher's exact test (non-parametric test), would reduce false positives because the numbers or variation of each biomarker positive samples is varied in an unbiased comprehensive bioinformatics analysis.

Panels D,E,F give p-values for the mutation or amplification status, which makes no sense. The authors should just determine what it is and give the numbers and mutations. Critically, none of these three panels supports the claim that MYC-amplified or mutated cancer cells are particularly sensitive to the inhibitor. For this, much stronger data would be necessary.

As described in A), in this revised manuscript, we re-analyzed the Oncopanel data using 169 cell lines, whose genomic data could be obtained from the CCLE database. Consequently, we now claim that *MYC* amplification is associated with the sensitivity to T-025 in solid cancer cell lines but not in hematological cancer cell lines. To support our claim, we showed that *MYC* amplified solid cancer cell lines were statistically sensitive to non *MYC* amplified solid cancer cell lines (Fig 3B). The names of cell lines from representative tissues such as breast, lung, colon, and CNS tissue and its *MYC* amplification status are shown in Fig EV5.

In the hematological cancer cell lines, increased sensitivity in *MYC* altered cell lines (amplified, driver mutated, or translocated *MYC*) was not observed (Fig EV2). The names of the hematological cancer cell lines and its *MYC* family gene status are also shown in Fig EV2.

In the description of this figure, the authors state: "Both *MYC* amplification and mutation lead to *MYC* activation": I do not think that this is consistent with the literature.

Thank you for your comment. We agree with your comment and have removed the sentence from the main text.

In this revised manuscript, we carefully checked the *MYC* gene alterations. Regarding mutations, we divided them into driver and passenger mutations using the OncoKB database.

The analysis suggests that several mutations can enhance sensitivity to these drugs, some with significantly lower p-values than *MYC*. It would be good to see one or two of the other predictions tested.

We thank the referee for this comment. We have checked the mutations of *ZFP106*, *FBN3*, *AUTS2*,

or *PCMI*, where *p*-values were lower than for *MYC*. However, the role of these mutations in cancer has not been demonstrated yet. The involvement of *CREBBP*, *CHD7*, *ROBO2*, or *ASXL1* in sensitivity to T-025 might be addressed in future publications. In light of this comment, however, we have added some thoughts about the relationship between sensitivity and mutations in *CREBBP* or *ROBO2* to the Discussion section of our revised manuscript.

Figure 6

Why is the colour code different from panel 2A?

This was an error and we have corrected the figure accordingly.

As before, it would be important to see immunoblots of some of the affected proteins to judge the significance and magnitude of the effects.

Immunoblots of *MYC* levels relative to endogenous *MYC* levels of some tumor cells should be shown to judge the magnitude of overexpression.

The immunoblotting results, including a comparison of endogenous *MYC* expression levels and ectopic expression levels in SK-MEL-28, are shown below (modified Fig 6E).

Although the ectopic expression of *MYC* was higher than the endogenous level in *MYC*-amplified cells, we showed that depletion of *MYC* by siRNA rendered cancer cells less sensitive to T-025 (Fig 5I), suggesting that physiological expression level of *MYC* affect sensitivity to T-025.

The authors should also perform a bioinformatics analysis to provide some insight which exons are skipped? Do they have a particular feature? Are they in genes of a specific functional category? How does exon skipping in these genes affect cell growth?

We performed bioinformatics analysis and found that genes with exons skipped by T-025 were enriched in DNA repair, cell cycle and RNA export pathways (Fig. 4F). The feature (e.g. motif) of the skipped exon caused by a CLK inhibitor (T3) has been investigated thoroughly and reported in a previous paper (Funnell et al, 2017). Since the skipped exons caused by T3 and those caused by T-025 largely overlapped (Fig EV4C), we did not include this bioinformatics analysis in this manuscript.

Reference

Araki S, Dairiki R, Nakayama Y, Murai A, Miyashita R, Iwatani M, Nomura T, Nakanishi O (2015) Inhibitors of CLK protein kinases suppress cell growth and induce apoptosis by modulating pre-mRNA splicing. *PLoS One* 10: e0116929

Funnell T, Tasaki S, Oloumi A, Araki S, Kong E, Yap D, Nakayama Y, Hughes CS, Cheng SG, Tozaki H, Iwatani M, Sasaki S, Ohashi T, Miyazaki T, Morishita N, Morishita D, Ogasawara-

Shimizu M, Ohori M, Nakao S, Karashima M, Sano M, Murai A, Nomura T, Uchiyama N, Kawamoto T, Hara R, Nakanishi O, Shumansky K, Rosner J, Wan A, McKinney S, Morin GB, Nakanishi A, Shah S, Toyoshiba H, Aparicio S (2017) CLK-dependent exon recognition and conjoined gene formation revealed with a novel small molecule inhibitor. *Nat Commun* **8**: 7

He X, Riceberg J, Soucy T, Koenig E, Minissale J, Gallery M, Bernard H, Yang X, Liao H, Rabino C, Shah P, Xega K, Yan ZH, Sintchak M, Bradley J, Xu H, Duffey M, England D, Mizutani H, Hu Z, Guo J, Chau R, Dick LR, Brownell JE, Newcomb J, Langston S, Lightcap ES, Bence N, Pulukuri SM (2017) Probing the roles of SUMOylation in cancer cell biology by using a selective SAE inhibitor. *Nat Chem Biol* **13**: 1164-1171

Hsu TY, Simon LM, Neill NJ, Marcotte R, Sayad A, Bland CS, Echeverria GV, Sun T, Kurley SJ, Tyagi S, Karlin KL, Dominguez-Vidana R, Hartman JD, Renwick A, Scorsone K, Bernardi RJ, Skinner SO, Jain A, Orellana M, Lagisetti C, Golding I, Jung SY, Neilson JR, Zhang XH, Cooper TA, Webb TR, Neel BG, Shaw CA, Westbrook TF (2015) The spliceosome is a therapeutic vulnerability in MYC-driven cancer. *Nature* **525**: 384-388

Kim E, Ilagan JO, Liang Y, Daubner GM, Lee SC, Ramakrishnan A, Li Y, Chung YR, Micol JB, Murphy ME, Cho H, Kim MK, Zebari AS, Aumann S, Park CY, Buonamici S, Smith PG, Deeg HJ, Lobry C, Aifantis I, Modis Y, Allain FH, Halene S, Bradley RK, Abdel-Wahab O (2015) SRSF2 Mutations Contribute to Myelodysplasia by Mutant-Specific Effects on Exon Recognition. *Cancer Cell* **27**: 617-630

Koh CM, Bezzi M, Low DH, Ang WX, Teo SX, Gay FP, Al-Haddawi M, Tan SY, Osato M, Sabo A, Amati B, Wee KB, Guccione E (2015) MYC regulates the core pre-mRNA splicing machinery as an essential step in lymphomagenesis. *Nature* **523**: 96-100

Lee SC, Dvinge H, Kim E, Cho H, Micol JB, Chung YR, Durham BH, Yoshimi A, Kim YJ, Thomas M, Lobry C, Chen CW, Pastore A, Taylor J, Wang X, Krivtsov A, Armstrong SA, Palacino J, Buonamici S, Smith PG, Bradley RK, Abdel-Wahab O (2016) Modulation of splicing catalysis for therapeutic targeting of leukemia with mutations in genes encoding spliceosomal proteins. *Nat Med* **22**: 672-678

Rodgers JT, Haas W, Gygi SP, Puigserver P (2010) Cdc2-like kinase 2 is an insulin-regulated suppressor of hepatic gluconeogenesis. *Cell Metab* **11**: 23-34

Zhou X, Li X, Cheng Y, Wu W, Xie Z, Xi Q, Han J, Wu G, Fang J, Feng Y (2014) BCLAF1 and its splicing regulator SRSF10 regulate the tumorigenic potential of colon cancer cells. *Nat Commun* **5**: 4581

2nd Editorial Decision

12 April 2018

Thank you for the submission of your revised manuscript to EMBO Molecular Medicine. We have now received the enclosed reports from the referees that were asked to re-assess it. As you will see the reviewers are now globally supportive and I am pleased to inform you that we will be able to accept your manuscript pending the following final amendments:

Please submit your revised manuscript within two weeks. I look forward to seeing a revised form of your manuscript as soon as possible.

I look forward to reading a new revised version of your manuscript as soon as possible.

***** Reviewer's comments *****

Referee #2 (Remarks for Author):

Iwai and colleagues have thoroughly updated their manuscript addressing most of my concerns. The authors support their initial hypothesis with new data and experiments that reinforce the relation

between MYC status and CLK2 expression in terms of T-025 sensitivity, although with significant heterogeneity depending on tumor type. Particularly, the new clinical data showing that breast cancer patients with high-expression of CLK2 and MYC amplification have worse outcome. MYC downregulation in MYC-amplified breast cancer cell lines (SK-BR3 and MCF-7) sensitizes cells to T-025 treatment independently of CLK2 status. Overall, these results suggest that although both CLK2 levels and MYC status are important factors for determining drug sensitivity, the effect that these two have in the response is not always accumulative and depends on tumor cell type.

Overall, the manuscript has significantly improved from the reviewing process.

Referee #3 (Comments on Novelty/Model System for Author):

Multiple cell line models and in vivo models used

Referee #3 (Remarks for Author):

The original version of this manuscript was substandard. The revised manuscript has been improved significantly. The emphasis of the revised manuscript on the link between MYC expression and response to T-025 makes this story quite interesting. That link was suggested in the first version, but that data was far from compelling. This MYC connection is now much more solid and suggests a biomarker-driven strategy to take this drug to the clinic.

Corresponding Author Name: Toshiyuki Nomura

Manuscript Number: EMM-2017-08289